# Novel approaches to estimating turbulent kinetic energy dissipation rate from low and moderate resolution velocity fluctuation time series

Marta Wacławczyk[1], Yong-Feng Ma[1], Jacek M. Kopeć[1,2], and Szymon P. Malinowski[1]

[1]Institute of Geophysics, Faculty of Physics, University of Warsaw, Warsaw, Poland
[2]Interdisciplinary Centre of Mathematical and Numerical Modelling, University of Warsaw, Warsaw, Poland

*Correspondence to:* marta.waclawczyk@igf.fuw.edu.pl

**Abstract.** In this paper we propose two approaches to estimating the turbulent kinetic energy dissipation rate, based on the zero-crossing method by Sreenivasan et al. [*J. Fluid Mech.*, **137**, 1983]. The original formulation requires a fine resolution of the measured signal, down to the smallest dissipative scales. However, due to finite sampling frequency, as well as measurement errors, velocity time series obtained from airborne experiments are characterized by the presence of effective spectral cut-offs. In contrast to the original formulation the new approaches are suitable for use with signals originating from airborne experiments. The suitability of the new approaches is tested using measurement data obtained during the Physics of Stratocumulus Top (POST) airborne research campaign as well as synthetic turbulence data. They appear useful and complementary to existing methods. We show the number-of-crossings based approaches respond differently to errors due to finite sampling and finite averaging than the classical power spectral method. Hence, their application for the case of short signals and small sampling frequencies is particularly interesting, as it can increase the robustness of turbulent kinetic energy dissipation rate retrieval.

## 1 Introduction

Despite the fact that turbulence is one of the key physical mechanisms responsible for many atmospheric phenomena, information on Turbulent Kinetic Energy (TKE) dissipation rate $\epsilon$ based on *in situ* airborne measurements is scarce. Research aircraft are often not equipped to measure wind fluctuations with spatial resolution better than few tens of meters (Wendisch and Brenguier, 2013). Due to various problems related to e.g. inhomogeneity of turbulence along the aircraft track and/or artifacts related to inevitable aerodynamic problems (Khelif et al., 1999; Kalgorios and Wang, 2002; Mallaun et al., 2015), estimates of $\epsilon$ at such low resolutions using power spectral density or structure functions are complex and far from being standardised (e.g. compare procedures in Strauss et al. (2015), Jen-La Plante et al. (2016)). The question arises: can we do any better? Or at least can we introduce alternative methods to increase robustness of $\epsilon$ retrievals?

In the literature, there exist several different methods to estimate $\epsilon$ using the measured velocity signal as a starting point. One of them is the zero- or threshold-crossing method (Sreenivasan et al., 1983) which, instead of calculating the energy spectrum or velocity structure functions, requires counting of the signal zero- or threshold crossing events, see Fig. 1a. Their mean number

per unit length is related to the turbulent kinetic energy dissipation rate. The zero-crossing method is based on a direct relation between $\epsilon$ and the root mean square of the velocity derivative (Pope, 2000), hence, the measured signal should be resolved down to the smallest scales. However, this is not achievable in the case of flight measurements with moderate time-resolutions. Using the Taylor's hypothesis, the measured time series can be converted into a spatial signal and the sampling frequency will

correspond to scales which are typically $2-3$ orders of magnitude larger than the Kolmogorov scales. As a result, the number of zero-crossings per unit length for such signal is much smaller than the one corresponding to a high resolution velocity signal where turbulence intensity is the same.

Interestingly, Kopeć et al. (2016) have shown, that the dissipation rates estimated from such $N_L$ using very low resolution signals, although underestimated, were proportional to $\epsilon$ calculated using structure functions scaling in the inertial range. In

the follow up analyses we found that this is also the case for moderate-resolution airborne data from different sources. This led us to a question whether it would be possible to modify the zero-crossing method such that it can also be applied to moderate- or low-resolution measurements whilst mitigating the observed underestimation at the same time. In this work we propose two possible modifications of the zero-crossing method. The first one is based on a successive filtering of a velocity signal and inertial range arguments. In the second approach we use an analytical model for the unresolved part of the spectrum and

calculate a correcting factor to $N_L$, such that the standard relation between $\epsilon$ and $N_L$ can be used.

The new approaches are tested on velocity signals obtained during the Physics of Stratocumulus Top (POST) research campaign, which was designed to investigate the marine stratocumulus clouds and the details of vertical structure of stratocumlus-topped boundary layer (STBL) (Gerber et al., 2013; Malinowski et al., 2013). The observed winds were measured using the CIRPAS Twin-Otter research aircraft with sampling frequency $f_s = 40Hz$, which corresponds to the resolution $2.75m$ for the

speed of the aircraft $55m/s$. Additional tests of the method with synthetic velocity signals as suggested by Frehlich et al. (2001) are also performed.

The present paper is structured as follows. In section 2 we review existing methods to estimate dissipation rate of the turbulent kinetic energy. Next, in Section 3 we propose the two modifications of the zero-crossing method. They are applied to a single signal from flight 13 and synthetic turbulence data and discussed in detail in Section 4. Next, in Section 5 we apply the

procedures to several data sets from flights 10 and 13 to show that the results of new approaches compare favourably with those obtained from standard power-spectrum and structure function methods. This is followed by Conclusions where the advantages of the new proposals and perspectives for further study are discussed.

## 2    Previous methods to retrieve the energy dissipation rate from measured velocity time series

The need to estimate the turbulent kinetic energy dissipation rate $\epsilon$ as well as variety of available data resulted in formulating a

number of estimation methods. Two of the most commonly used approaches are the power spectral density and the structure-function approach. Both are based on the inertial range arguments, which follow from the Kolmogorov's second similarity hypothesis (Kolmogorov, 1941), hence, they are also called "indirect methods" (Albertson et al., 1997). With the assumption of local isotropy the one-dimensional longitudinal and transverse wavenumber spectra in the inertial range are given by (Monin

and Yaglom, 1975; Pope, 2000):

$$E_{11}(k_1) = C_1 \epsilon^{2/3} k_1^{-5/3}, \quad E_{22}(k_1) = C_1' \epsilon^{2/3} k_1^{-5/3}. \tag{1}$$

Here $k_1$ is the longitudinal component of the wavenumber vector $\mathbf{k} = (k_1, k_2, k_3)$, $C_1 \approx 0.49$ and $C_1' \approx 0.65$ if $k_1$ units are rad/m (cf. Pope (2000), Eqs. [6.242,6.243]). $E_{11}$ is related to the energy-spectrum function $E(k)$

$$E_{11}(k_1) = \int\limits_{k_1}^{\infty} \frac{E(k)}{k} \left( 1 - \frac{k_1^2}{k^2} \right) \mathrm{d}k, \tag{2}$$

here $k = |\mathbf{k}|$. As discussed in Pope (2000) experimental data confirm Eqs. (1) within $20\%$ of the predicted values of $C_1$ and $C_1'$ over two decades of wavenumbers. Within the validity of the local isotropy assumption of Kolmogorov (1941), the energy-spectrum function can be approximated by the formula (Pope, 2000):

$$E(k) = C \epsilon^{2/3} k^{-5/3} f_L(kL) f_\eta(k\eta), \tag{3}$$

here $C \approx 1.5$ as supported by experimental data, $f_L$ and $f_\eta$ are non-dimensional functions, which specify the shape of energy-spectrum in, respectively, the energy-containing and the dissipation range. $L = k^{3/2}/\epsilon$ denotes the length scale of large eddies and $\eta = (\nu^3/\epsilon)^{1/4}$ is the Kolmogorov length scale connected with the dissipative scales (Pope, 2000), where $\nu$ is the kinematic viscosity. The function $f_L$ tends to unity for large $kL$ whereas $f_\eta$ tends to unity for small $k\eta$, such that in the inertial range the formula $E(k) = C \epsilon^{2/3} k^{-5/3}$ is recovered.

Within the validity of the Taylor's hypothesis Eq. (1) can be converted to the frequency spectra, where $k_1 = (2\pi f)/U$ and $U$ is the magnitude of the vector difference between the aircraft velocity and the wind velocity, i.e. the true air speed. The vector difference is averaged along the displacement which defines $k_1$. The frequency $f$ is measured in $1/\mathrm{s}$, $U$ in m/s and $k_1$ in rad/m. In order to estimate the dissipation rate from the atmospheric turbulence measurements, several assumptions should be taken. Most importantly, one assumes that the turbulence is homogeneous and isotropic and that the inertial range scaling Eqs. (1) holds. Then, frequency spectrum of the longitudinal velocity component in the inertial range is (e.g., Oncley et al., 1996; Siebert et al., 2006):

$$S(f) = C_1 \left( \frac{U}{2\pi} \right)^{2/3} \epsilon^{2/3} f^{-5/3}. \tag{4}$$

The value of a constant $C_1 \approx 0.49$ used in this work is related to the one-sided spectra. Hence, by $E_{11}$, $E_{22}$ or $S(f)$ we denote the one-sided spectra, which, integrated over argument from 0 to $\infty$ yield the variance of the signal. With Eq. (4), the turbulent kinetic energy dissipation rate can be estimated from the power spectral density (PSD) of the measured signal.

Alternatively, one can consider the $n$-th order longitudinal structure functions $D_n = \langle (u_L(x+r,t) - u_L(x,t))^n \rangle$, here $u_L$ is the longitudinal component of velocity and $r$ is a displacement along the direction defined by $u_L$. In the inertial subrange, the second and third-order structure functions are related to the dissipation rate $\epsilon$ by the formulas (Pope, 2000):

$$D_2(r) = C_2 \epsilon^{2/3} r^{2/3}, \quad D_3(r) = -\frac{4}{5} \epsilon r. \tag{5}$$

Experimental results of Saddoughi and Veeravalli (1994) indicate that $C_2 \approx 2$. with an accuracy of $\pm 15\%$.

Another method, also based on the formula (3) is the velocity variance method (Fairall et al., 1980; Bouniol et al., 2004; O'Connor et al., 2010). Let us consider a homogeneous velocity field, converted to time series $u(t)$ with the use of Taylor's hypothesis. The mean-square value of this signal $\langle u^2(t) \rangle = u^{'2}$ is equal to the integral form 0 to $\infty$ of the one-sided power spectral density $S(f)$ over the frequency space.

The signal $u(t)$ is next filtered with a band-pass filter with cut-off numbers $[f_{low}, f_{up}]$ in the frequency space. Assuming that the filter is perfect, i.e. it is a rectangle in the frequency space, after the filtering a signal $u_f(t)$ with the variance

$$u_f^{'2} = \int_{f_{low}}^{f_{up}} S(f) df \tag{6}$$

is obtained. The above formula represents the portion of kinetic energy of $u(t)$ contained in the frequencies between $f_{low}$ and $f_{up}$. Fairall et al. (1980); Bouniol et al. (2004); O'Connor et al. (2010) substitute Eq. (3) for $S(f)$ into (6) and integrate to obtain the following formula for the dissipation rate:

$$\epsilon = \left[ \frac{2(2\pi)^{2/3} u_f^{'2}}{3 C_1 U^{2/3} (f_{low}^{-2/3} - f_{up}^{-2/3})} \right]^{3/2}. \tag{7}$$

Yet another method, also used in the atmospheric turbulence analysis (Sreenivasan et al., 1983; Poggi and Katul, 2009, 2010; Wilson, 1995; Yee et al., 1995), is based on the number of zero- or level-crossings of the measured velocity signal. It dates back to the early work of Rice (1945) who considered a stochastic processes $q$ and its derivative with respect to time $\partial q / \partial t$. He then assumed that these two processes have Gaussian statistics and are independent. The formulation of this method results from investigating how frequently the signal crosses the level zero $q(t) = 0$, see Fig. 1a. Working under those assumptions Rice (1945) showed that the number of up-crossings of the zero level per unit time is:

$$N^2 = \frac{\langle (\partial q / \partial t)^2 \rangle}{4\pi^2 \langle q^2 \rangle}. \tag{8}$$

As $\langle (\partial q / \partial t)^2 \rangle$ is proportional to the dissipation rate of the kinetic energy, the zero-crossing method can be used to estimate this quantity. As it was argued by Sreenivasan et al. (1983), Eq. (8) holds also with less restricted assumptions, with only $q$ having Gaussian statistics and, moreover, even for strongly non-Gaussian velocity signals the number of zero-crossings was close to the theoretical value from Eq. (8). For a spatially varying signal, Eq. (8) can be expressed as follows, using the characteristic wavenumber $k_c$ and the one-sided wavenumber spectra (He and Yuan, 2001):

$$k_c = \sqrt{\frac{\int_0^\infty k_1^2 E_{11} dk_1}{\int_0^\infty E_{11} dk_1}}. \tag{9}$$

The characteristic wavelength is equal to $\lambda_c = 2\pi / k_c$. Hence, the mean number of crossings (up- and downcrossings) per unit length $N_L$, with, on average, two crossing per $\lambda_c$ is

$$N_L = \frac{2}{\lambda_c} = \frac{1}{\pi} k_c. \tag{10}$$

We will now introduce the two-point longitudinal correlation of velocity $R_{ij}(r_1\mathbf{e}_1) = \langle u_i(\mathbf{x},t)u_j(\mathbf{x}+r_1\mathbf{e}_1,t)\rangle$, where $\mathbf{e}_1$ is the standard basis vector and assume that the flow is statistically stationary and homogeneous and statistics do not depend either on time $t$ or point $\mathbf{x}$.

5    Using the inverse Fourier transform, the 11 component of the two-point correlation tensor $R_{11}$ and its derivatives can be written in terms of $E_{11}$ as follows (Pope, 2000):

$$R_{11}(r_1\mathbf{e}_1) = \int_0^\infty E_{11}(k_1)\cos(k_1 r_1)\mathrm{d}k_1, \quad R_{11}''(r_1\mathbf{e}_1) = -\int_0^\infty E_{11}(k_1)k_1^2\cos(k_1 r_1)\mathrm{d}k_1, \tag{11}$$

where $R_{11}''$ denotes the second-order derivative of $R_{11}$. With those relationships we can rewrite Eq. (9) in the following manner:

$$10 \quad k_c = \sqrt{\frac{\int_0^\infty k_1^2 E_{11}(k_1)\mathrm{d}k_1}{\int_0^\infty E_{11}(k_1)\mathrm{d}k_1}} = \sqrt{\frac{-R_{11}''(0)}{R_{11}(0)}}. \tag{12}$$

We further define the Taylor longitudinal microscale $\lambda_f$ with the use of $R_{11}''(0)$ and $R_{11}(0)$

$$\lambda_f = \left(-\frac{1}{2}\frac{R_{11}''(0)}{R_{11}(0)}\right)^{-1/2}. \tag{13}$$

Hence, Eq. (10) implies that the number of crossings per unit length is related to the longitudinal Taylor's microscale $\lambda_f$ through

$$15 \quad \lambda_f = \frac{\sqrt{2}}{\pi}\frac{1}{N_L} \quad \Longrightarrow \quad \frac{1}{\lambda_f^2} = \frac{1}{2}\pi^2 N_L^2. \tag{14}$$

Relations (11–14) are valid for any statistically homogeneous vector fields, regardless of whether or not they are isotropic (Monin and Yaglom, 1975), provided that $k_c$ is the characteristic wavenumber along the longitudinal direction. However, homogeneity alone is not a sufficient assumption to estimate the TKE dissipation rate $\epsilon$ of a $3D$ turbulent field from velocity signals measured along the $1D$ aircraft flight path (Chamecki and Dias, 2004). We further use the local isotropy assumption to

20   write a relation between dissipation and the Taylor microscales (Pope, 2000)

$$\epsilon = \frac{30\nu u'^2}{\lambda_f^2} = \frac{15\nu u'^2}{\lambda_g^2}, \tag{15}$$

where $\lambda_g = \lambda_f/\sqrt{2}$ is the Taylor transverse microscale. Hence, finally, substituting Eq. (14) into Eq. (15) we obtain (Poggi and Katul, 2010)

$$\epsilon = 15\pi^2\nu u'^2 N_L^2. \tag{16}$$

25   For the transverse velocity time series Eq. (16) has a factor $7.5$ instead of $15$.

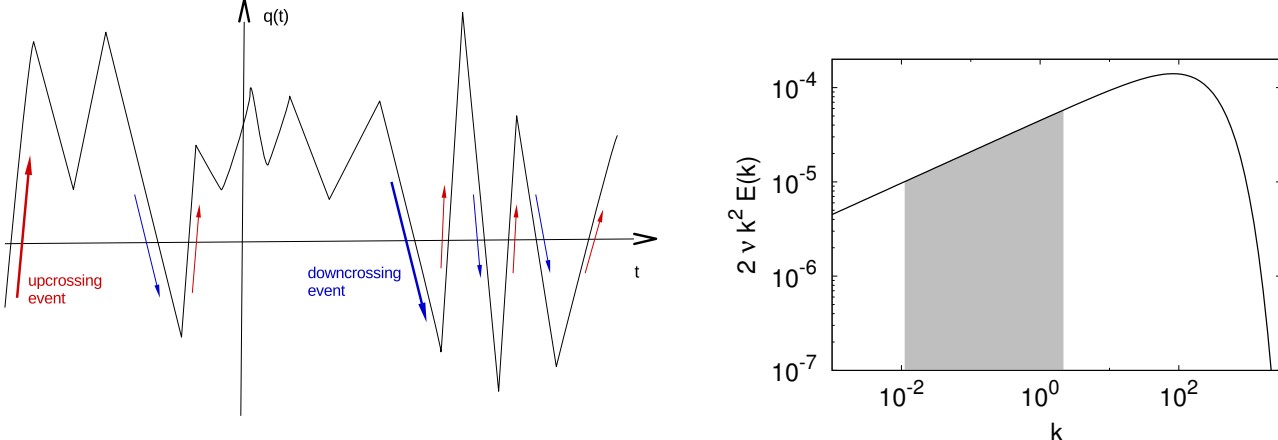

**Figure 1.** a) A signal $q(t)$ crossing the level $q = 0$. b) Dissipation spectra: the range of $k$-numbers covered by the POST measurements is denoted by the colour shading.

## 3 New proposals to estimate dissipation rate from a velocity signal with a truncated high-frequency part of the energy spectrum

Based on Eqs. (9) and (10) it is clear that the number of zero-crossings is related to the 11 component of the dissipation tensor $D_{11}(k) = 2\nu k^2 E_{11}(k)$:

$$5 \quad \pi^2 u'^2 N_L^2 = \int_0^\infty k^2 E_{11} \mathrm{d}k. \tag{17}$$

Figure 1b presents the profile of $D(k) = 2\nu k^2 E(k)$ where $E(k)$ is described by the model spectrum (3) with $f_\eta = \exp(-\beta k \eta)$ (Pope, 2000), here $\beta = 2.1$ and $\eta = 2mm$. It is seen that the large wavenumber (small scale) part of the spectrum has the most significant impact on the resulting value of $N_L$.

At the same time the data available from the POST measurements can only account for a small part of the total dissipation
10  spectrum, shown qualitatively as a shaded region in Fig. 1b. The lower bound of this region follows from a finite size of the averaging window while the upper is related to the finite Nyquist frequency which equals 20Hz for the POST measurements.

If one was to use the zero-crossing method (Eq. 16) in order to estimate $\epsilon$ it is clear that the measured number of signal zero-crossings would lead to significant underestimation of the spectrum integral as compared to the full spectrum measurements down to the very small scales. We would like to propose reformulation of the original zero-crossing method in order to estimate
15  the dissipation rate from the number of signal zero-crossings based on a restricted range of $k$-values available from the airborne measurements. Two proposals for such procedures are given further in the article.

## 3.1 Method based on successive filtering of a signal

Let us consider a signal $u_1(t)$ resolved in a certain range of frequencies $f_0 < f < f_1$. Converting the wavenumber spectrum to the frequency spectrum we obtain from Eq. (17) the following relation for the number of signal-crossings per unit time

$$u_1'^2 N_1^2 = 4 \int_{f_0}^{f_1} f^2 S(f) \mathrm{d}f. \tag{18}$$

5    Similarly as in the velocity variance method described in Section 2, let us now filter the signal using a band-pass filter characterized by a different cut-off frequency $f_2 < f_1$. In such a case we obtain a different signal $u_2(t)$ with a reduced number of zero-crossings $N_2 < N_1$:

$$u_2'^2 N_2^2 = 4 \int_{f_0}^{f_2} f^2 S(f) \mathrm{d}f. \tag{19}$$

If we subtract Eq. (19) from Eq. (4.2) we obtain

$$10 \quad u_1'^2 N_1^2 - u_2'^2 N_2^2 = 4 \int_{f_2}^{f_1} f^2 S(f) \mathrm{d}f. \tag{20}$$

In the inertial range $S(f)$ is described by Eq. (4), hence, if both $f_1$ and $f_2$ belong to the inertial range

$$u_1'^2 N_1^2 - u_2'^2 N_2^2 = 4 C_1 \left( \frac{U}{2\pi} \right)^{2/3} \epsilon^{2/3} \int_{f_2}^{f_1} f^{1/3} \mathrm{d}f = 3 C_1 \left( \frac{U}{2\pi} \right)^{2/3} \epsilon^{2/3} \left( f_1^{4/3} - f_2^{4/3} \right). \tag{21}$$

If we proceed further and filter the signal using a series of cut-off frequencies $f_i < f_2$, we can estimate $\epsilon$ from Eq. (21) using a linear least squares fitting method.

15    In the above derivation we assumed a perfect filter, rectangular in the frequency space is used. The issue of frequency response characteristics of a filter will be discussed further in Section 4.1.

## 3.2 Method based on recovering the missing part of the spectrum

In this method we attempt to account for the impact of the missing part of the dissipation spectrum by introducing a correcting factor to the number of zero-crossings per unit length $N_L$. The number of crossings per unit length is calculated from the

20    measured signal where the fine-scale fluctuations having the highest wavenumber $k_{cut}$ will be denoted by $N_{cut}$ and the variance of this signal will be denoted by $u_{cut}'^2$. From Eq. (17) it follows that $N_{cut}$ is related to $N_L$ by the formula

$$u'^2 N_L^2 = u_{cut}'^2 N_{cut}^2 \frac{\int_0^\infty k_1^2 E_{11} \mathrm{d}k_1}{\int_0^{k_{cut}} k_1^2 E_{11} \mathrm{d}k_1} = u_{cut}'^2 N_{cut}^2 \left( 1 + \frac{\int_{k_{cut}}^\infty k_1^2 E_{11} \mathrm{d}k_1}{\int_0^{k_{cut}} k_1^2 E_{11} \mathrm{d}k_1} \right). \tag{22}$$

We then assume a certain form of the energy spectrum, Eq. (3). For simplicity we take $f_L = 1$, i.e. we neglect the contribution of largest scales to the value of the dissipation rate based on zero-crossings and we will consider two different forms of $f_\eta$, as

proposed in Pope (2000). First being a simple exponential form

$$f_\eta = e^{-\beta k \eta}, \tag{23}$$

with $\beta = 2.1$ and a second, more complex formula

$$f_\eta = e^{\left\{-\left[(\beta k \eta)^4 + (\beta c_\eta)^4\right]^{1/4} + \beta c_\eta\right\}}, \tag{24}$$

here $\beta = 5.2$ and $c_\eta = 0.4$. With this, the energy spectrum reads

$$E(k) = C\epsilon^{2/3} k^{-5/3} f_\eta(\beta k \eta), \tag{25}$$

here $C = 1.5$. The integral from 0 to $\infty$ of the dissipation spectrum $2\nu k^2 E(k)$ should be equal to $\epsilon$, which results in $\beta = 2.1$ in Eq. (23) and provides a relation between $\beta$ and $c_\eta$ in Eq. (24). The latter case, due to the additional degree of freedom in $f_\eta$ fits the experimental data better in the dissipative range (Pope, 2000).

The corresponding one-dimensional spectrum $E_{11}$ can be calculated from Eq. (2)

$$E_{11}(k_1) = C\epsilon^{2/3} \int_{k_1}^{\infty} k^{-8/3} f_\eta(\beta k \eta) \left(1 - \frac{k_1^2}{k^2}\right) dk. \tag{26}$$

Next we change the variables in the integral Eq. (26) to $\xi = \beta k \eta$, introduce Eq. (26) into Eq. (22) and once again change the variables to $\xi_1 = \beta k_1 \eta$. As a result we obtain

$$u'^2 N_L^2 \approx u_{cut}'^2 N_{cut}^2 \underbrace{\left[1 + \frac{\int_{k_{cut}\beta\eta}^{\infty} \xi_1^2 \int_{\xi_1}^{\infty} \xi^{-8/3} f_\eta(\xi)\left(1 - \frac{\xi_1^2}{\xi^2}\right) d\xi d\xi_1}{\int_0^{k_{cut}\beta\eta} \xi_1^2 \int_{\xi_1}^{\infty} \xi^{-8/3} f_\eta(\xi)\left(1 - \frac{\xi_1^2}{\xi^2}\right) d\xi d\xi_1}\right]}_{\mathcal{C}_\mathcal{F}} = u_{cut}'^2 N_{cut}^2 \mathcal{C}_\mathcal{F}, \tag{27}$$

here $\mathcal{C}_\mathcal{F}$ is the correcting factor. The value of $\epsilon$ can be calculated numerically using an iterative procedure.

As a starting point for this procedure, a first guess for the Kolmogorov length $\eta = (\nu^3/\epsilon)^{1/4}$ should be given. With this, we calculate the correcting factor $\mathcal{C}_\mathcal{F}$ from Eq. (27) taking either the form Eq. (23) or (24) for $f_\eta$. Next, from Eq. (16) the value of dissipation can be estimated as

$$\epsilon = 15\pi^2 \nu u_{cut}'^2 N_{cut}^2 \mathcal{C}_\mathcal{F}. \tag{28}$$

We start the next iteration by calculating again the Kolmogorov length $\eta = (\nu^3/\epsilon)^{1/4}$, the corrected value of $\mathcal{C}_\mathcal{F}$ from Eq. (27) and the new value of $\epsilon$ from Eq. (28). After several iterations the procedure converges to the final values of the dissipation rate and Kolmogorov's length $\eta$ with an error defined by a prescribed norm $\Delta\eta = |\eta^{n+1} - \eta^n| \leq d_\eta$. The successive steps are summarized in a form of Algorithm 1.

It should be noted that in this approach we do not have the empirical inertial range constant $C$, and we calculate the dissipation rate directly from the formula with viscosity, Eq. (28), as in the original zero-crossing method see Eq. (16) and Poggi and Katul (2010).

**Algorithm 1** Procedure of iterative $\epsilon$ determination based on missing spectrum part recovery

$\quad \epsilon \leftarrow 15\pi^2\nu u^{'2} N_{cut}^2$

$\quad \eta \leftarrow (\nu^3/\epsilon)^{1/4}$

$\quad \Delta\eta \leftarrow 100 d_\eta$

$\quad$ **while** $\Delta\eta > d_\eta$ **do**

$\quad\quad$ Use Eq. (27) to calculate $\mathcal{C}_{\mathcal{F}}$

$\quad\quad \epsilon \leftarrow 15\pi^2\nu u^{'2} N_{cut}^2 \mathcal{C}_{\mathcal{F}}$

$\quad\quad \Delta\eta \leftarrow |\eta - (\nu^3/\epsilon)^{1/4}|$

$\quad\quad \eta \leftarrow (\nu^3/\epsilon)^{1/4}$

$\quad$ **end while**

## 4 In depth analysis of the proposed methods' behaviour

### 4.1 Method based on the number of zero-crossings of successively filtered signal

In order to present the more detailed properties of the procedure we used velocity signal from one of the horizontal flight segments that took place within the turbulent atmospheric boundary layer. This segment was a part of flight 13 of the POST airborne research campaign (Gerber et al., 2013; Malinowski et al., 2013). The data were provided in the East, North, Up (ENU) coordinate system. For further study we have calculated time series of the longitudinal velocity component along the track. The signals sampling frequency was $f_s = 40\,\text{Hz}$ and the duration was $T = 438.75\,\text{s}$. The magnitude of the vector difference between the aircraft velocity and the wind velocity $U$, averaged over track vector was about $55\,\text{ms}^{-1}$ and the standard deviation $u' = 0.28\text{ms}^{-1}$.

We have estimated the dissipation rate based on the number of zero-crossings, according to the methods outlined in section 3.1. The average dissipation rate calculated from the frequency spectrum and the structure function for the whole flight fragment Eqs. (4) and (5) was close to equal, respectively, $\epsilon_{PSD} = 2.48\times10^{-4}\,\text{m}^2\text{s}^{-3}$ and $\epsilon_{SF} = 2.52\times10^{-4}\,\text{m}^2\text{s}^{-3}$. These values were obtained from the linear least-squared fit procedure in the range $f = 0.3-5\,\text{Hz}$ for the frequency spectrum and $r = 11-183\,\text{m}$ for the structure function, see Fig. 2.

Before applying the threshold crossing procedures the signal had to be filtered in order to eliminate errors due to large scale tendencies as well as small scale measurements noise. For this purpose we used the sixth order low-pass Butterworth filter (Butterworth, 1930) implemented in Matlab ®. Figure 3 presents the velocity signal over $t = 50$s before filtering (top graph) and the same signal after filtering with $f_{cut} = 5\,\text{Hz}$ and $f_{cut} = 1\,\text{Hz}$.

The original formula of Rice (1945) was derived for the case when both the signal and its derivative have Gaussian probability density functions (PDF's) and are statistically independent. In general, such assumptions does not hold for turbulent signals. However, as discussed by Sreenivasan et al. (1983), experimental observations and further theoretical studies indicate that the formula of Rice (1945) has a more general applicability than it was mathematically proven for and is satisfied with a fair accuracy even for the case of strongly non-Gaussian signals. Figure 4a presents PDF's of the normalised original signal and the

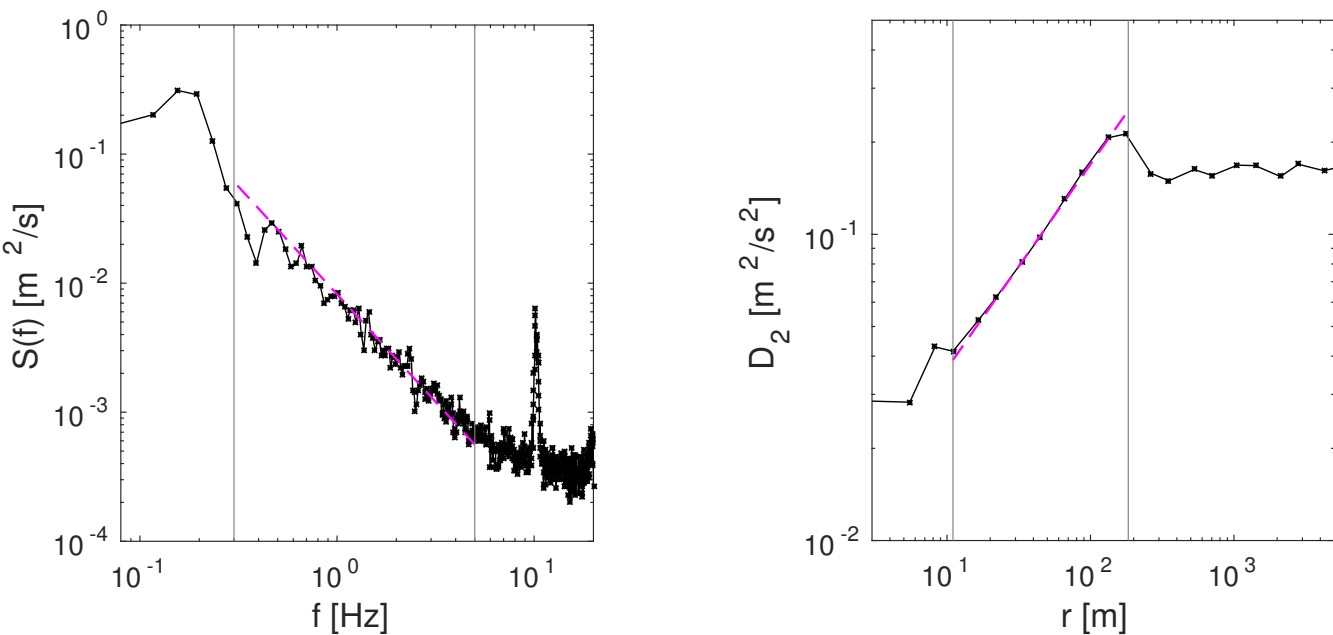

**Figure 2.** a) Frequency spectrum of the measured signal (POST), b) second order structure function. Polynomial fit is presented as a coloured dashed line.

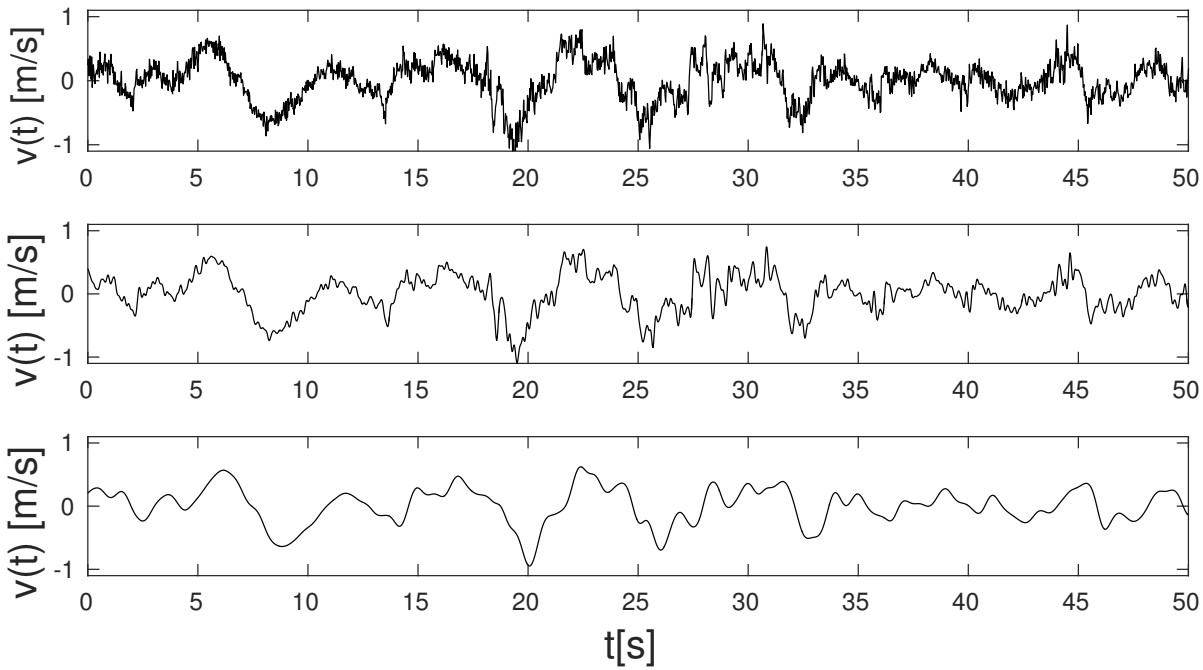

**Figure 3.** Measured velocity fluctuations: top graph - unfiltered signal, middle graph - signal filtered with $f_{cut} = 5\,\mathrm{Hz}$, bottom graph - signal filtered with $f_{cut} = 1\,\mathrm{Hz}$.

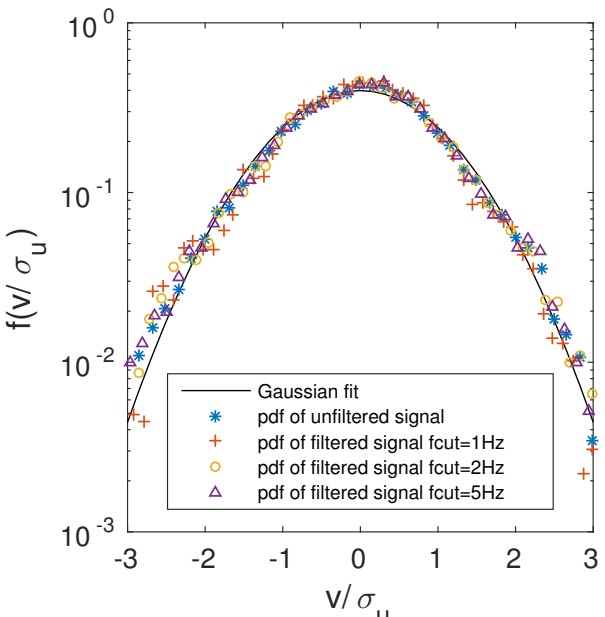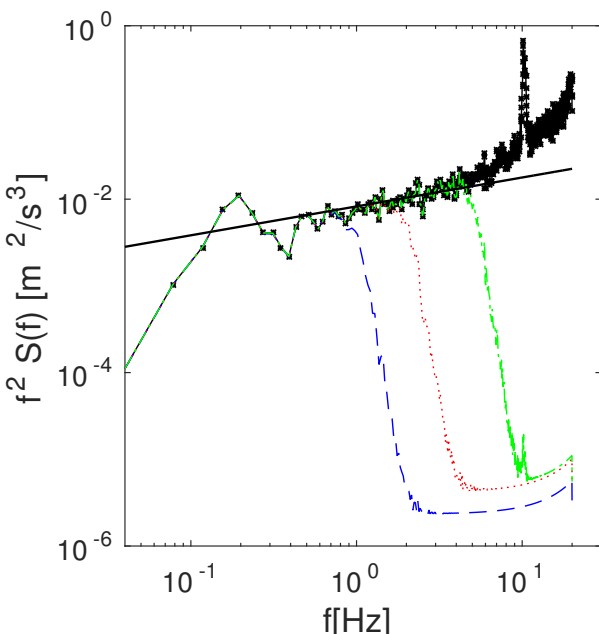

**Figure 4.** a) PDF's of the normalised unfiltered and filtered measured signals compared with the normalised Gaussian curve. b) Spectra $f^2 S(f)$ of the unfiltered signal (black line with symbols), signal filtered with $f_{cut} = 5\,\text{Hz}$ (green, solid line), signal filtered with $f_{cut} = 2\,\text{Hz}$ (red dotted line), signal filtered with $f_{cut} = 1\,\text{Hz}$ (blue, dashed line).

filtered signals compared with the normalised Gaussian distribution. As it is seen, filtering does not lead to significant changes in the investigated PDF's.

It is worth noting that the spectra ($f^2 S(f)$, Fig. 4b) display a peak at $f = 10 Hz$. This phenomenon has been indicated in the previous analyses of POST (Jen-La Plante et al., 2016) and appears due to measurement errors. However, as the highest cut-off frequencies used in the present study are 5 Hz, it should not affect our results.

In order to use the method based on successive signal filtering we filtered the signal with different values of $f_{cut}$ in the range $f_{cut} = 0.1 - 19\,\text{Hz}$. For each $f_{cut} = f_i$ we calculated the number of zero-crossings $N_i$ based on the filtered signal. The zero-crossing event was detected when the product of two consecutive values of velocity fluctuation $v(t)v(t + \Delta t) < 0$, here $\Delta t = 1/f_s = 0.025\,\text{s}$. In order to estimate the value of dissipation rate we used Eq. (21). The values $u_i^{'2}$ were calculated from filtered time series. Results for $f_1 = 0.3\,\text{Hz}$ and $f_i$ in the range $(0.3\,\text{Hz}, 5\,\text{Hz})$ are presented in Fig. 5. Using Eq. (21) we have used linear fitting of the differences $u_i^{'2} N_i^2 - u_1^{'2} N_1^2$ against $f_i^{4/3} - f_1^{4/3}$. The resulting value for the analysed flight section was $\epsilon_{NCF} = 2.54 \times 10^{-4}\,\text{m}^2\text{s}^{-3}$. This value is comparable with the estimations performed using classic methods based on the power spectra and structure functions.

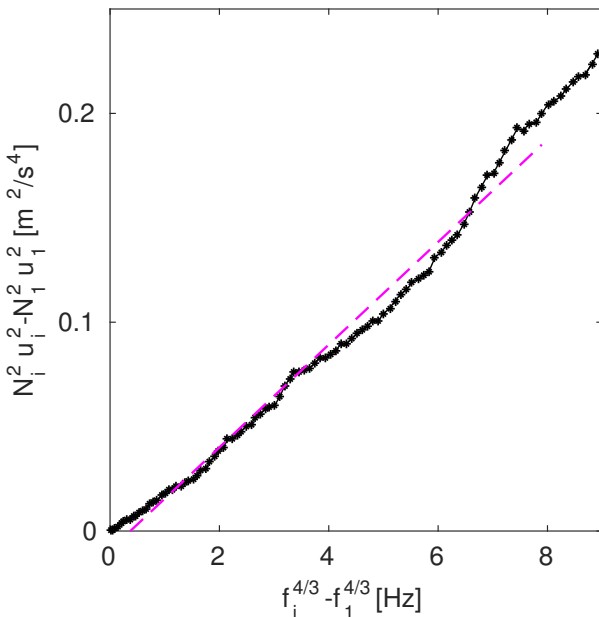

**Figure 5.** Scaling of $N_i^2 u_i^2$ with filter cut-off $f_{cut}$ calculated for the measured signal (POST). The linear fit from formula (21) is given by the magenta dashed line.

## 4.2 Simulation analysis and error estimates

Even if the local isotropy assumption of Kolmogorov (1941) is satisfied with a good accuracy, the TKE dissipation rate estimates are subject to errors that can result from a finite sampling frequency of a signal, a finite time window, sensor bias and noise. The last of those three causes was investigated in Sreenivasan et al. (1983), where it was shown that both the variance of the noise $\langle n^2 \rangle$ as well as variance of its derivative $\langle \dot{n}^2 \rangle$ influence the measured number of crossings. A possible remedy was proposed by Poggi and Katul (2010) who suggested to use the threshold-crossings, i.e. counting the number of times a signal crosses a given threshold $T \neq 0$, instead of the zero-crossings in case of signals with low signal-to-noise ratios. As for the signal considered in the previous section the signal-to-noise ratio becomes significant at higher frequencies (above $5Hz$), see Fig. 2, which are removed by the low-pass filter used in the proposed number of crossings method. We also applied the method of Poggi and Katul (2010), however, as it did not lead to any systematic change of our estimates we further present results for the zero-crossings only.

In order to quantify the errors resulting from the finite sampling frequency and finite time window and test the performance of the proposed method we performed the simulation analysis (Frehlich et al., 2001; Sharman et al., 2014). Results will be compared with the standard spectral retrieval estimates, without any additional corrections. However, we note in passing that spectral methods can be improved to account for the bias errors. The example is the maximum likelihood approach (Sharman et al., 2014) where, instead of the von Kármán model, a model power-spectral density $S_f^{model}$ is used, which takes into account

the procedure for generating the empirical spectrum from discrete time series of finite length. Analogous approaches could also be formulated for the methods based on the number of crossing, which is a perspective for a further study.

To test the performance of new proposals we generated a number of artificial velocity signals with frequency spectra and two point correlation functions prescribed by the von Kármán (1948) model. The equations resulting from applying this model to the one-sided spectra considered in this paper are written below.

$$R_{11}(r_1\mathbf{e}_1) \approx 0.592548\, u'^2 \left(\frac{r}{L_0}\right)^{1/3} K_{1/3}\left(\frac{r}{L_0}\right), \qquad S(f) \approx 0.475448\, \frac{2\pi}{U} \frac{u'^2 L_0}{\left[1 + L_0^2 \left(\frac{2\pi f}{U}\right)^2\right]^{5/6}}, \tag{29}$$

here $K_{1/3}$ is the modified Bessel function of order $1/3$. Coefficients of the Fourier series expansion of velocity signal were calculated as

$$w_j = \sqrt{W_j}(a + ib) \tag{30}$$

here $i = \sqrt{-1}$, $a$ and $b$ are random numbers from the standard Gaussian distribution with zero mean and unit variance and $W_j = S(f_j)\Delta f$, $j = 1, \ldots, N$. Alternatively, the coefficients $W_j$ can be calculated as the discrete Fourier transform of $R_{11}$, as described in Frehlich et al. (2001). The artificial velocity signal is finally constructed as the discrete inverse Fourier transform of $w_j$, see Frehlich et al. (2001).

We used artificial signals with $U = 55\,\mathrm{ms}^{-1}$ and the standard deviation $u' = 0.28\mathrm{ms}^{-1}$. Those characteristics correspond to the ones of the signal considered in the previous Section 4.1. We set $L_0 = 83.9$ in Eq. (29) to obtain also a comparable dissipation rate estimate $\epsilon = 2.5 \cdot 10^{-4}\mathrm{m}^2\mathrm{s}^{-3}$. Our first aim was to test how a finite sampling rate influences the number of crossings. For this purpose in each run we created an artificial signal of length $N = 2^{17}$ points and with the sampling frequency 200Hz (five times larger as the sampling of the signal considered in Section 4.1), which resulted in signal duration $t \approx 650\mathrm{s}$. We treated this velocity series as a "reference". Next, we took every fifth sample of this signal to create a 40Hz velocity time series. We then calculated the number of crossings, as described in Section 4.1 and the power spectral density. We repeated the procedure 500 times and calculated average of the obtained profiles, see Fig. 6. Due to the finite sampling frequency we observe the effect of aliasing - spectral densities for $f$ higher than the Nyquist frequency are added to the spectral densities at $f < 20Hz$. Distortions are visible for higher frequencies both in the power spectrum, Fig. 6a, as well as $N_i^2 u_i^2$ profiles, Fig. 6b. We estimated the TKE dissipation rate from the averaged profiles, using the method described in Section 4.1, Eq. (21), keeping the lower bound of the fitting range $f_1 = 0.3$Hz constant and changing the upper bound $f_2$ from 1 to 19Hz. Results are presented in Fig. 7 and compared with the corresponding $\epsilon_{PSD}$ values, using the von Kármán model as a reference model spectrum.

We observe an increase of $\epsilon_{PSD}$ estimates with increasing $f_2$ and a moderate increase of $\epsilon_{NCF}$ over the input $\epsilon = 2.5 \cdot 10^{-4}\mathrm{m}^2\mathrm{s}^{-3}$, which shows that the number of crossings respond to finite sampling effects differently than the power spectrum. We note here that $\epsilon_{NCF}$ calculated from the averaged profiles of 200Hz "reference" signal (black line in Fig. 7) seem to be slightly overpredicted in comparison to the input $\epsilon$ , especially for smaller $f_2$. A possible reason is the response of a filter used in the number of crossings method. We attempted to estimate this error using relation () between the number of crossings

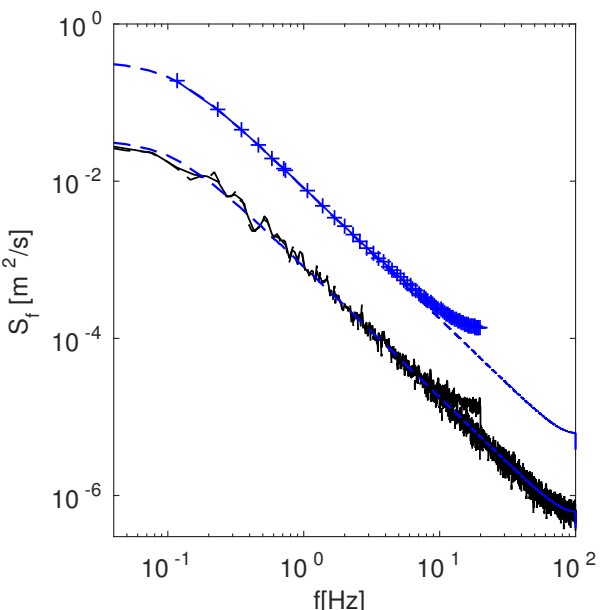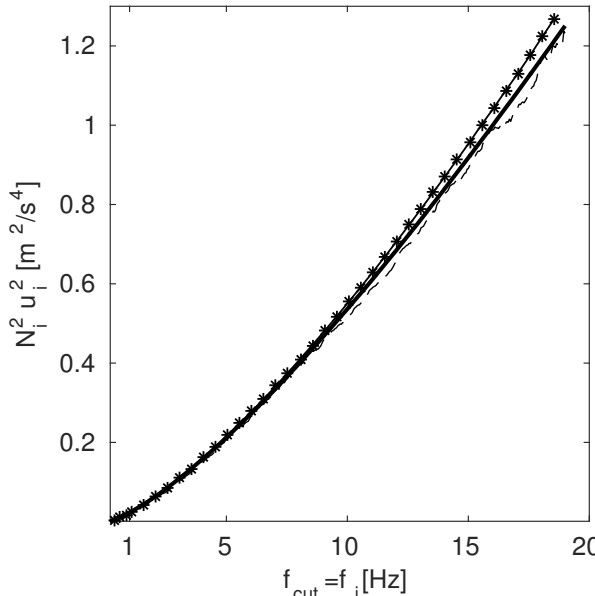

**Figure 6.** a) Mean $S(f)$ profiles calculated from the simulation analysis: blue dashed lines - synthetic signal sampled with 200Hz, blue line with symbols - synthetic signal sampled with 40Hz, black lines - profiles from a single signal with $u'^2 = 0.0885 \text{ms}^{-1}$, b) corresponding averaged $N_i^2 u_i^2$ profiles: solid line - 200Hz signal, line with symbols - 40Hz signal, thin black line - profile from a single signal with $u'^2 = 0.28 \text{ms}^{-1}$.

and the dissipation spectrum. We first integrated $f^2 S_f$ of an unfiltered signal from $f_0$ to $f_i$. Next, we calculated a spectrum $S_f^{filtered}$ of a band-pass filtered signal taking $f_1$ and $f_i$ as the lower and upper bounds of a filter. We integrated $f^2 S_f^{filtered}$ over the whole available range of $f$. The difference between the two integrals should represent a "correction" due to filter response. The $\epsilon$ values estimated from the corrected $N_i^2 u_i'^2$ are presented in Fig. 7 as a black dot-dashed line. As it is seen the estimations for the lower $f_2$ are improved, however, as $f_2$ increases, $\epsilon_{NCF}$ seem to be underpredicted. A possible reason for

5  this might be that the filter influence the number of crossings statistics somewhat differently than the spectrum alone.

Next, we tested the influence of the finite temporal window on the calculated statistics. We generated 1000 artificial signals, each time changing slightly the $u'$ value in Eq. (29) which led to a change of input $\epsilon$, see Sharman et al. (2014), the value of $L_0$ remained unchanged. For each signal we estimated $\epsilon_{PSD}$ from the standard power spectral density using the Welch's overlapped segment averaging estimator implemented in Matlab ®with a $2^8$ window and $\epsilon_{NCF}$ from the number of crossings,

10  Eq. (21). We did this tests for the 40Hz signals and the fitting range $0.3 - 5$Hz. We first decreased the time window, taking each time only $1/8$ of the created artificial signal for the analysis, which, in terms of $L_0$ from Eq. (29) resulted in the signal length equal approximately $L \approx 50 L_0$. Results of $\epsilon_{PSD}$ and $\epsilon_{NCF}$ estimates as functions of corresponding input $\epsilon$ from the theoretical profile Eq. (29) are presented in Fig. 8 (upper plots). It can be seen that the bias error is larger for $\epsilon_{PSD}$, however,

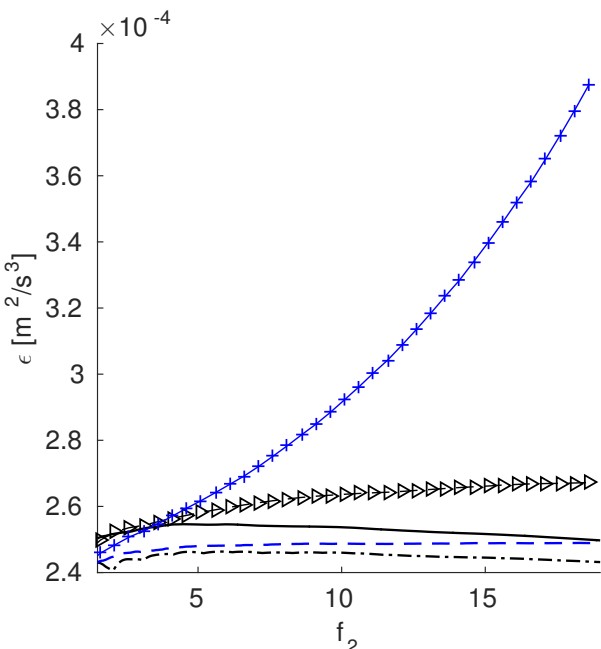

**Figure 7.** Values of the dissipation rate from simulation analysis as a function of higher value of the fitting range $f_2$ estimated based on the averaged profiles from Fig. 6 of: $S(f)$ blue dashed line - synthetic 200Hz signal, blue line with $+$ symbols - 40Hz synthetic signal; $N_i^2 u_i^2$, Eq. (21), solid line - 200Hz synthetic signal, line with $\triangleright$ symbols - 40Hz synthetic signal; $N_i^2 u_i^2$ for 200Hz signal with the filter response correction - black dot-dashed line. The input $\epsilon = 2.5 \cdot 10^{-4} \mathrm{m}^2 \mathrm{s}^{-3}$.

the scatter of $\epsilon_{NCF}$ is larger. The linear fits and the correlation coefficients are

$$
\begin{aligned}
\epsilon_{PSD} &= 0.9104\,\epsilon - 2.32 \cdot 10^{-5}, \quad r = 0.9898, \\
\epsilon_{NCF} &= 0.9878\,\epsilon + 6.80 \cdot 10^{-5}, \quad r = 0.9343.
\end{aligned}
\tag{31}
$$

We repeated the simulation analysis for signals with $2^{17}$ points, i.e. with $L \approx 400 L_0$ obtaining

$$
\begin{aligned}
\epsilon_{PSD} &= 1.0377\,\epsilon + 4.56 \cdot 10^{-6}, \quad r = 0.9898, \\
\epsilon_{NCF} &= 1.0379\,\epsilon + 2.25 \cdot 10^{-5}, \quad r = 0.9989.
\end{aligned}
\tag{32}
$$

Hence, for the signal length comparable to the lengths from the POST campaign we can expect a small underprediction of $\epsilon_{PSD}$ estimates due to bias error and some overprediction due to aliasing, see Fig. 7. Both result in a small overprediction of $\epsilon_{PSD}$ (Fig. 8, left column, lower plot). As far as $\epsilon_{NCF}$ is concerned, the simulation analysis shows that it is less sensitive to the bias error (Fig. 8, right column), however it has a larger scatter than $\epsilon_{PSD}$, at least for the generated artificial velocity fields. Results for the 40Hz signal are slightly overpredicted (Fig. 8, right column, lower plot) due to aliasing and the fact that the number of crossing method gives somewhat larger $\epsilon$ estimates in this fitting range, see Fig. 7.

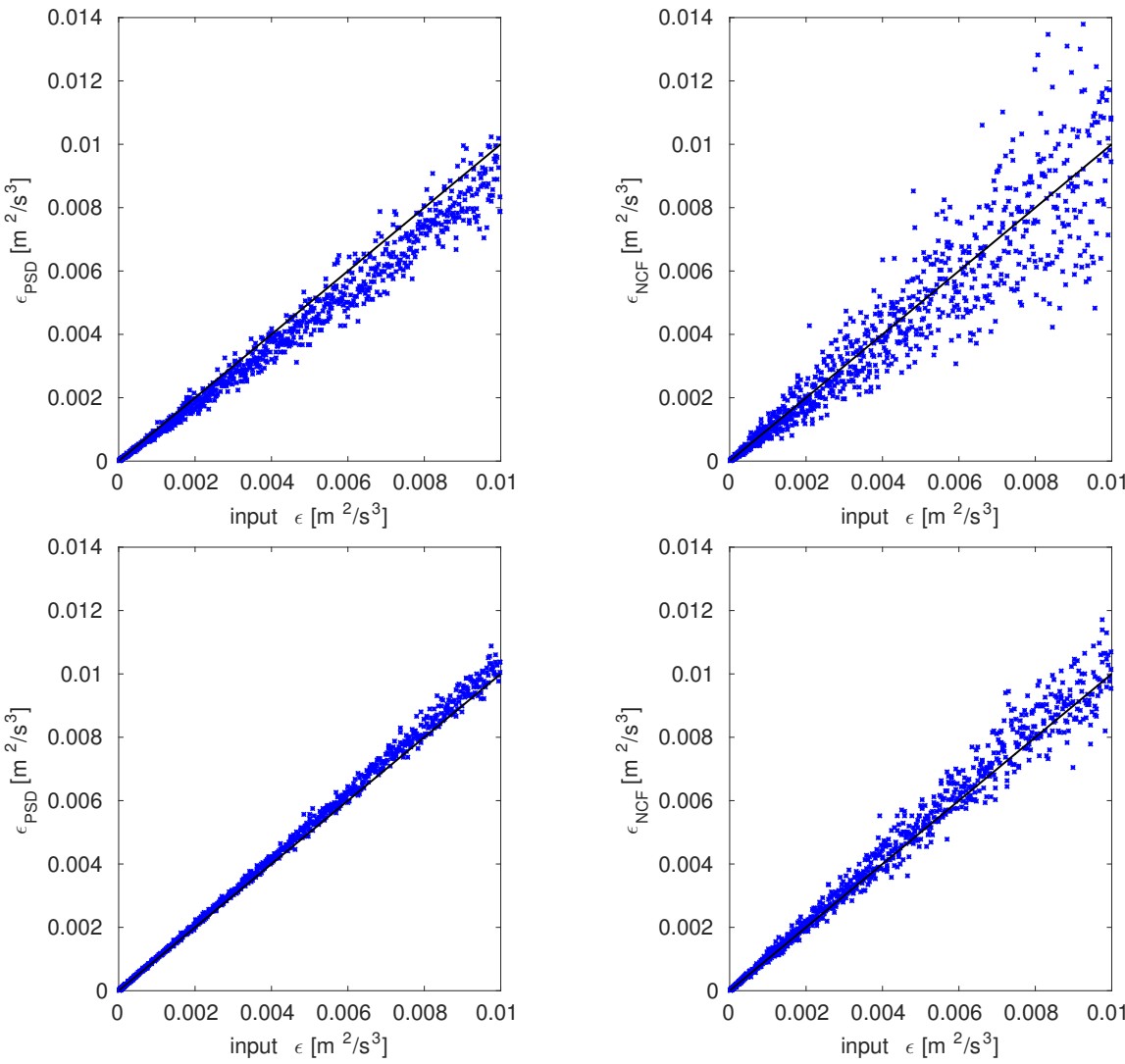

**Figure 8.** Estimated values of $\epsilon_{PSD}$ and $\epsilon_{NCF}$ for synthetic 40Hz signals and fitting range $0.3 - 5$Hz as functions of corresponding input $\epsilon$ resulting from the theoretical profile, Eq. (29) for upper plots: signals with $L \approx 50L_0$, lower plots: signals with $L \approx 400L_0$.

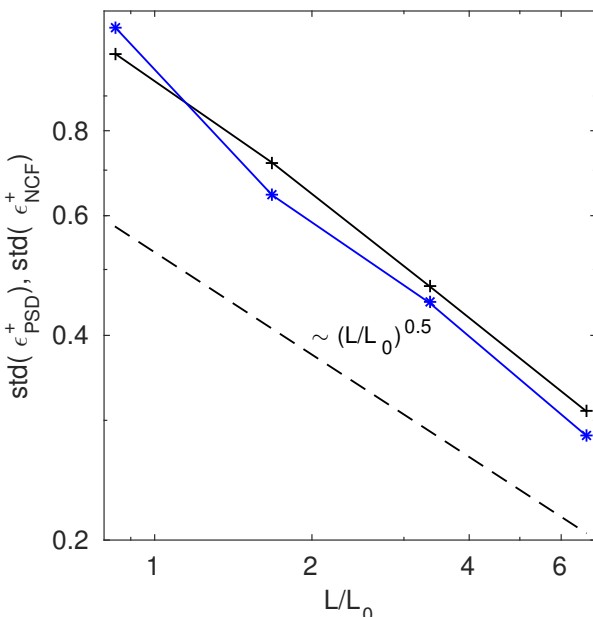

**Figure 9.** TKE dissipation rate estimates from simulation analysis for synthetic signals sampled with 200Hz with $u' = 0.28\mathrm{ms}^{-1}$ normalised by the input epsilon $\epsilon = 2.5 \cdot 10^{-4} \mathrm{m}^2\mathrm{s}^{-3}$. Black lines with $+$ symbols - $\epsilon_{NCF}$, blue lines with $*$ symbols - $\epsilon_{PSD}$

Finally, we would like to address the issue of larger scatter observed for $\epsilon_{NCF}$. It follows from our study that the scatter in $\epsilon_{NCF}$ depends on the value of filter cut-offs in the fitting range. In the final test, we set $u' = 0.28\mathrm{ms}^{-1}$ and input epsilon $\epsilon = 2.5 \cdot 10^{-4} \mathrm{m}^2\mathrm{s}^{-3}$ and in constant and repeated the simulation 500 times for consecutively, $1/512, 1/265, 1/128, 1/65$ of the original signal of $2^{17}$ points, which, in terms of $L_0$ in Eq. (29) corresponds to approximately $1L_0, 2L_0, 4L_0, 6L_0$. We band-
5   pass filtered the signal and consider small fitting range of $16 - 18$Hz. We normalised the obtained results by the input $\epsilon$ and calculated their standard deviations. Results presented in Fig. 9 show that at least for this case the standard deviations of $\epsilon_{NCF}^{+}$ is comparable with the standard deviation of $\epsilon_{PSD}^{+}$

### 4.3 Method based on missing spectrum recovery

The measurement signal used in Section 4.1 was also analysed using the second method proposed in Section 3.2, Eqs. (27,28).
10   We will consider both formulas for the function $f_\eta$, Eqs. (23) and (24). The advantage of the simpler, exponential formula (23) is that the one-dimensional spectrum function $E_{11}(k_1)$, Eq. (26) can be written in terms of the incomplete $\Gamma$ function as follows

$$E_{11}(k_1) = C\epsilon^{2/3} (\beta\eta)^{5/3} \left[ \Gamma(-5/3, k_1\beta\eta) - (\beta\eta)^2 k_1^2 \Gamma(-11/3, k_1\beta\eta) \right], \tag{33}$$

here

$$\Gamma(a,x) = \int_x^\infty e^{-t}t^{a-1}\mathrm{d}t. \tag{34}$$

The correcting factor (27) in terms of the $\Gamma$ functions reads

$$\mathcal{C}_\mathcal{F} = 1 + \frac{\int_{k_{cut}\beta\eta}^\infty \xi_1^2 \left[\Gamma(-5/3,\xi_1) - \xi_1^2\Gamma(-11/3,\xi_1)\right]\mathrm{d}\xi_1}{\int_0^{k_{cut}\beta\eta} \xi_1^2 \left[\Gamma(-5/3,\xi_1) - \xi_1^2\Gamma(-11/3,\xi_1)\right]\mathrm{d}\xi_1}. \tag{35}$$

If Eq. (24) is used as a model for $f_\eta$, both integrals in Eq. (27) must be calculated numerically. On the other hand, as discussed in Pope (2000), (24) provides a better fit of experimental data in the dissipative range.

With such preparation we applied the iterative procedure, as described in Section 3.2. In POST experiment the effective cut off frequency was estimated at $f_{cut} = 5\,\mathrm{Hz}$ which corresponds to $k_{cut} = (2\pi f)/U = 0.57\,\mathrm{m}^{-1}$. Using the sixth order Butterworth filter this resulted in $u'^2 N_{cut}^2 = 0.0000719 \cdot 1/s^2$ for this signal. Accordingly we used the Algorithm 1 with $\nu = 1.5\cdot10^{-5}\mathrm{m}^2\mathrm{s}^{-1}$ and $d_\eta = 10^{-6}\mathrm{m}$. We approximated the integrals in Eq. (35) using the trapezoid rule. The results of successive approximations of $\mathcal{C}_\mathcal{F}$ and $\epsilon$ converge fast to a fixed value, independently of the initial guess of $\epsilon = \epsilon_0$ (Fig. 10a). The increment $dk_1$ in Eq. (35) was approximated by $\Delta k_1 = 5\cdot10^{-6}\,\mathrm{m}^{-1}$. For such choice we obtained $\epsilon_{NCR} = 2.61\cdot10^{-4}\,\mathrm{m}^2\mathrm{s}^{-3}$. We used this as a reference value. In order to estimate the numerical accuracy of the proposed algorithm we calculated the error $\Delta\epsilon = |\epsilon - \epsilon_{NCR}|$ for different values of $\Delta k_1$, see Fig. 10b. We obtain $\Delta\epsilon \sim \Delta k_1^{1.3}$.

Next we considered Eq. (24) as a model for $f_\eta$ and calculated the double integral in equation (27) using the trapezoid rule. We obtained the corresponding value $\epsilon_{NCR} = 2.58\cdot10^{-4}\,\mathrm{m}^2\mathrm{s}^{-3}$, which is very close to the estimate from the simple exponential form Eq. (23) and Eq. (35).

It is worth noting that the proposed method is accounting for a dominant (and not directly measured) part of the spectrum based on the theoretical knowledge about its shape. This knowledge is simply reduced to the form of the correcting factor $\mathcal{C}_\mathcal{F}$, Eq. (27), which contains integral of $k_1^2 E_{11}(k_1)$. Fig. 11 illustrates the relation between the measured and the estimated part of the spectrum for the analysed case with both forms of the function $f_\eta$, Eqs. (23) and (24). The spectral cut-off of the data considered here (5Hz) is in the inertial range, where $k_1^2 E_{11}(k_1)$ with both forms of $f_\eta$ functions are almost indistinguishable, see Fig. 11. At the same time integrals of the remaining (recovered) parts of $k_1^2 E_{11}(k_1)$ are almost equal, as independently of the choice of $f_\eta$, both dissipative spectra $2\nu k^2 E(k)$ must integrate to $\epsilon$. As a result, for the given spectral cut-off, $\epsilon_{NCR}$ estimates with the simple exponential Eq. (23) and Eq. (24) forms of $f_\eta$ are very close. This might change for larger cut-off frequencies. We expect that in case the cut-off frequency is placed in a region influenced by the form of $f_\eta$ function, the spectrum with Eq. (24) will provide better estimates of the TKE dissipation rate.

The result of application of this method $\epsilon_{NCR} = 2.58\cdot10^{-4}\,\mathrm{m}^2\mathrm{s}^{-3}$ with $f_\eta$ described by Eq. (24) $\epsilon_{NCR} = 2.61\cdot10^{-4}\,\mathrm{m}^2\mathrm{s}^{-3}$ with $f_\eta$ from Eq. (23) is comparable with the dissipation rates obtained using other methods, as discussed in Section 3.1, $\epsilon_{PSD} = 2.48\cdot10^{-4}\,\mathrm{m}^2\mathrm{s}^{-3}$, $\epsilon_{SF} = 2.52\cdot10^{-4}\,\mathrm{m}^2\mathrm{s}^{-3}$ and $\epsilon_{NCF} = 2.54\cdot10^{-4}\,\mathrm{m}^2\mathrm{s}^{-3}$. The relative differences between those estimations are less than 5%.

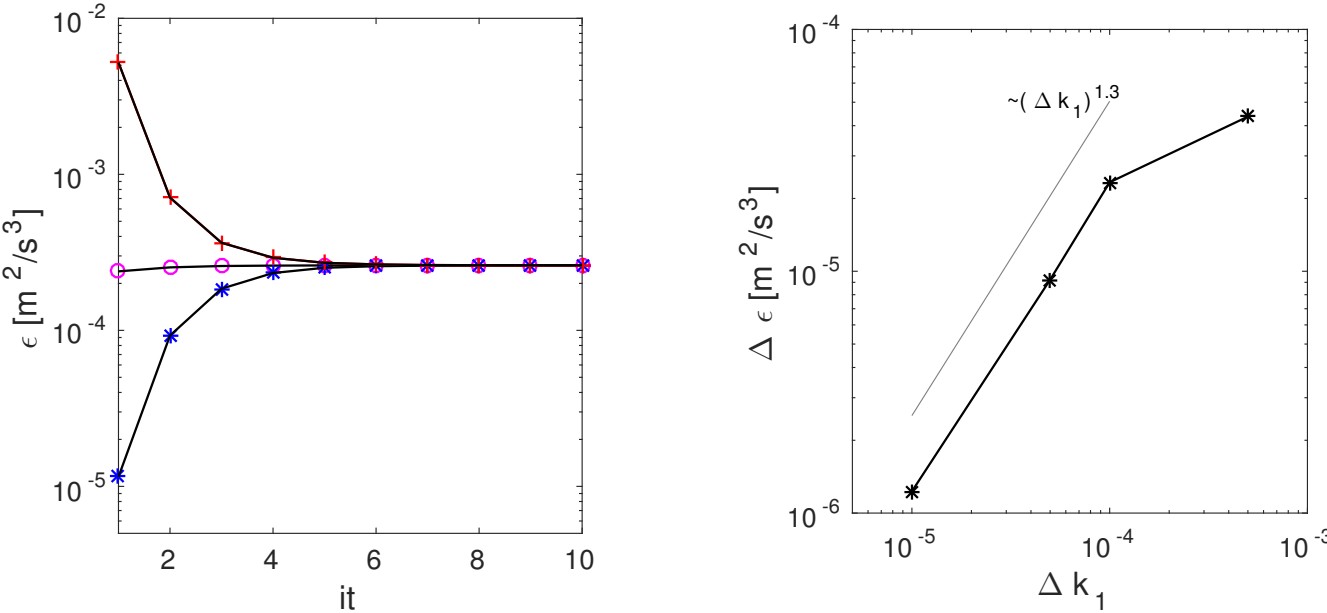

**Figure 10.** a) Values of $\epsilon$ calculated during the iterative procedure for different initial guesses of $\epsilon_0$. b) Error of $\epsilon$ as a function of $\Delta k$. The reference value is $\epsilon$ calculated with $\Delta k = 5 \cdot 10^{-6}\,\mathrm{m}^{-1}$.

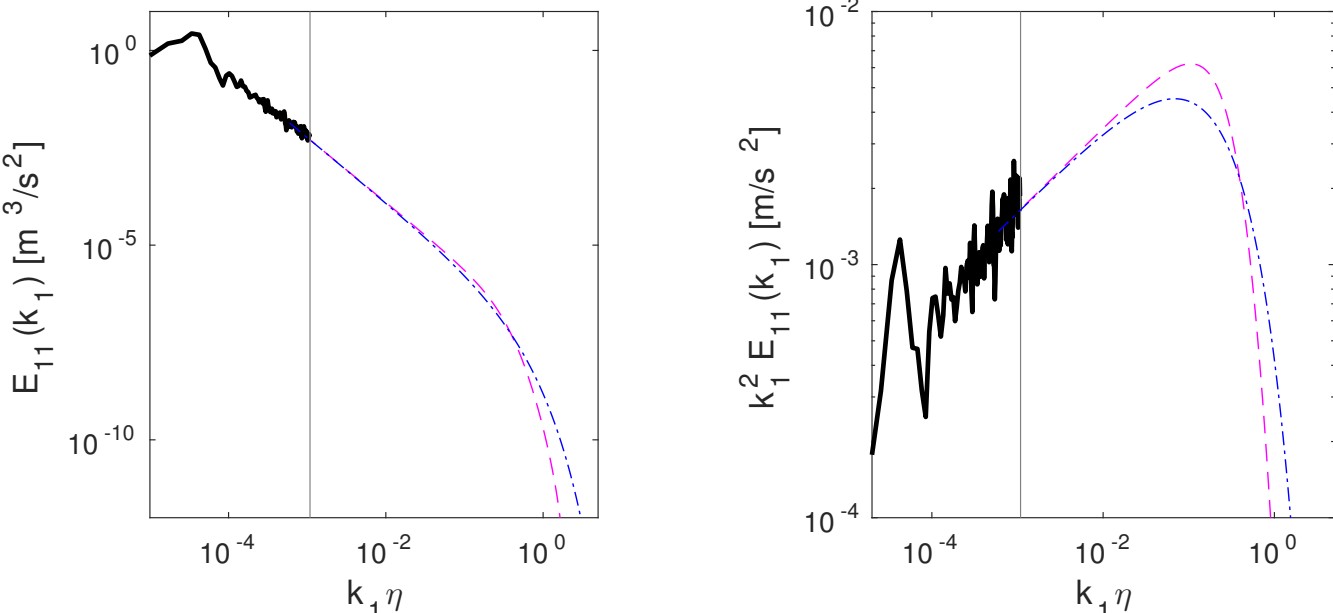

**Figure 11.** One-dimensional spectra: black solid line - measured part, dashed magenta line - recovered part with $f_\eta$ described by Eq. (23), dot-dashed blue line - recovered part with $f_\eta$ described by Eq. (24), a) energy spectrum $E_{11}(k_1)$, b) $k_1^2 E_{11}(k_1)$.

We finally checked estimates of the second method using synthetic signals as described in Section 4.2. For the cut-off 5Hz 500 artificial signals or length $L \approx 400 L_0$ and with input $\epsilon = 2.5 \cdot 10^{-4} \mathrm{m}^2 \mathrm{s}^{-3}$, resulting in the mean $\langle \epsilon_{NCR} \rangle = 2.55 \cdot 10^{-4} \mathrm{m}^2 \mathrm{s}^{-3}$ and a standard deviation equal 9% of the input $\epsilon$ value.

## 5    Broader overview of the methods' performance

Following the findings presented in the previous section both proposed methods were tested on much larger collection of data. For this purpose we used velocity signals also obtained during the POST research campaign. We have chosen horizontal segments at various levels within the boundary layer from flights $TO10$ and $TO13$. These flights were investigated in detail by Malinowski et al. (2013), due to the fact that they represent two thermodynamically and microphysically different types of stratocumulus topped bondary layer.

The dissipation rates of turbulent kinetic energy estimated from the standard structure function method $\epsilon_{SF}$ and dissipation rates estimated from the modified zero-crossing methods $\epsilon_{NCF}$ and $\epsilon_{NCR}$ introduced in Sections 3.1 and 3.2, respectively, are compared with the results obtained from the spectral method $\epsilon_{PSD}$ in Fig. 12. The use of simple exponential form of $f_\eta$, Eq. (23), or Eq. (24) did not lead to any visible change of results in Fig. 12. For flight 10 we obtained the following linear fits and the correlation coefficients $r$

$$
\begin{aligned}
\epsilon_{SF} &= 0.74\,\epsilon_{PSD} + 9.1 \cdot 10^{-5}, & r &= 0.997, \\
\epsilon_{NCF} &= 0.88\,\epsilon_{PSD} + 1.2 \cdot 10^{-5}, & r &= 0.995, \\
\epsilon_{NCR} &= 0.89\,\epsilon_{PSD} + 2.9 \cdot 10^{-5}, & r &= 0.999,
\end{aligned}
$$

while for flight 13 we have

$$
\begin{aligned}
\epsilon_{SF} &= 0.76\,\epsilon_{PSD} + 1.4 \cdot 10^{-4}, & r &= 0.956, \\
\epsilon_{NCF} &= 0.75\,\epsilon_{PSD} + 1.2 \cdot 10^{-4}, & r &= 0.881, \\
\epsilon_{NCR} &= 0.79\,\epsilon_{PSD} + 1.0 \cdot 10^{-4}, & r &= 0.987.
\end{aligned}
$$

The methods based on the signal zero-crossings give comparable results to those resulting from standard methods, in spite of the fact that the second method is based on different physical arguments (assumes form of the whole spectrum, including the dissipative range of frequencies). We believe that the there is a fair consistency in those results because one should take into account that the standard frequency spectra and structure function methods calculate approximate values of $\epsilon$. Moreover, we have indicated in Section 2 that the constants $C_1$ and $C_2$ in Eqs. (4) and (5) are estimated with an accuracy of $\pm 15\%$.

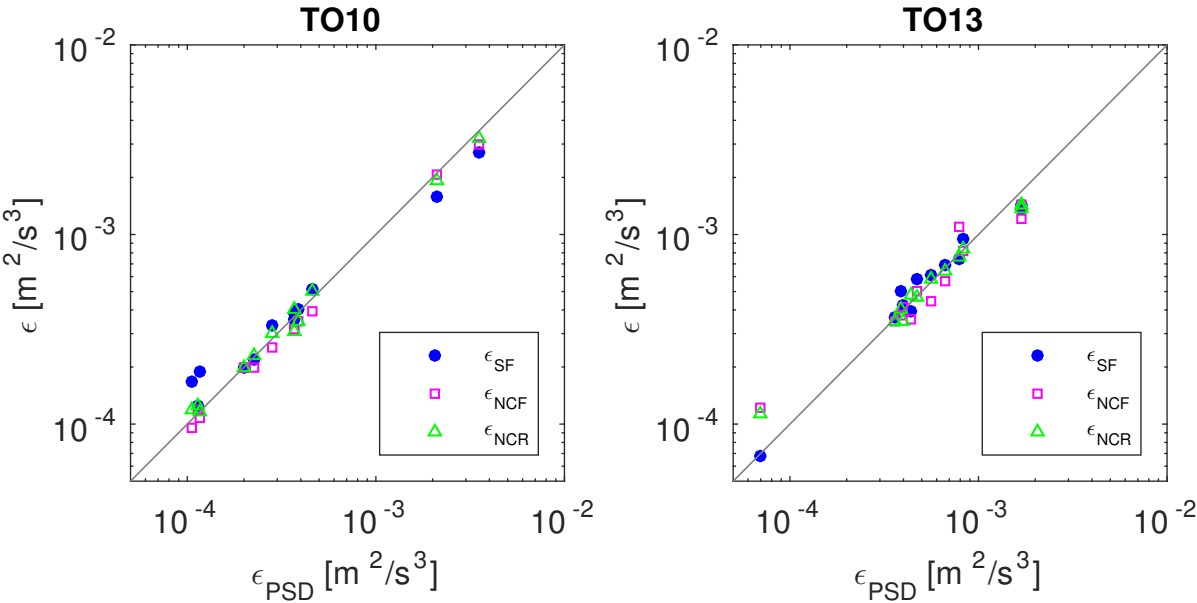

**Figure 12.** Dissipation rate of the kinetic energy estimated from the structure function method $\epsilon_{SF}$, zero-crossings of successively filtered signals $\epsilon_{NCF}$ and zero-crossings of signals with recovered part of the spectrum $\epsilon_{NCR}$ as a function of $\epsilon_{PSD}$ (from power spectra method). Each point represents an estimate from a single horizontal segment of flight in the atmospheric boundary layer, a) flight 10, b) flight 13.

## 6 Conclusions

In the present work we proposed two novel modifications of the zero-crossing method, such that it can be applied to moderate-resolution measurements. Turbulent kinetic energy dissipation rates obtained using the proposed methods shows fair agreement with results of the standard power-spectrum and structure function approaches.

We note that the standard structure function and power spectra methods are often used simultaneously, for better $\epsilon$ estimates (Chamecki and Dias, 2004), in spite of the same underlying physical arguments (second similarity hypothesis of Kolmogorov (1941)). Here, the proposed approach offers yet another option. Additionally, the second method with the spectrum recovery is based on different physical arguments, as it additionally makes use of the Kolmogorov's first similarity hypothesis and a model for the dissipation range of the spectrum. Still, it can be used for signals with spectral cut-offs, hence it offers an alternative to the spectral retrieval methods.

From the perspective of practical applications we can think of several possible advantages of the zero-crossing methods. First, the number of signal zero-crossings can be calculated without difficulty and the proposed procedures are easy to implement. Other advantages follow from the results of the simulation analysis performed in Section 4.2. For the created artificial velocity signals, the $\epsilon_{NCF}$ estimates responded differently to errors due to finite sampling or finite time windows than $\epsilon_{PSD}$. These differences in errors of the number of crossing and the power spectral method can make the former an additional tool to improve estimates from the atmospheric measurements. Here, a further, detailed study of bias assessment and removal is needed.

30    Moreover, we argue that the number of crossings method applied to the fully-resolved signals has become a fairly standard tool for $\epsilon$ estimates, used also in the atmospheric measurements, see e.g. Poggi and Katul (2010). Therein, the discussed advantages of the method are that no measurements of the signal gradients (to calculate the Taylor microscale) are required, no assumptions about scaling laws in structure functions (and power spectra) are needed and no simplifications in the TKE budget are adopted (for which $\epsilon$ is computed as a residual). The method proposed in the current manuscript, in particular, the second approach based on the recovered part of the spectrum, generalises number of crossing method and makes it applicable also for signals with spectral cut-off. Of course, on an additional cost, as certain form of the energy spectrum must be assumed in order to calculate the correcting factor $\mathcal{C}_{\mathcal{F}}$. Still, the proposed method can be interesting in particular for data with cut-offs reaching

5    the dissipation range, but still with part of this range missing (or contaminated with noise). In such case, using only the inertial range estimates may lead to a significant loss of information, as the data from the dissipation range are not taken into account. Finally, we can deal with a situation when the recorded amplitude of certain frequencies is deteriorated due to measurement errors still, the counted number of signal zero-crossings could remain unaffected. In such cases the zero-crossing method could be advantageous over the power-spectrum and structure-function methods.

There are several perspectives for further work. First, the proposed methods could be tested for a wider range of signals (e.g. from Eulerian measurements within the boundary layer adopting Taylor hypothesis), characterized by different resolutions and

5    obtained under varying atmospheric conditions, to assess the scope of their applicability. Second, as far as the model spectrum is concerned, comparison with fully-resolved experimental signals or Direct Numerical Simulations data would be valuable to test different forms of the model spectra from Pope (2000) or Bershadskii (2016).

## 7    Code availability

The MATLAB code written for the purpose of this study is available from the authors upon request.

10   ## 8    Data availability

POST data are available in the open database: https://www.eol.ucar.edu/projects/post/

*Acknowledgements.* We acknowledge financial support of the National Science Centre, Poland: MW through the project 2014/15/B/ST8/00180, Y-FM and SPM through the project 2013/08/A/ST10/00291. The POST field campaign was supported by US National Science Foundation through grant ATM-0735121 and by the Polish Ministry of Science and Higher Education through grant 186/W-POST/2008/0.

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
