# Peer review of "Novel approaches to estimating turbulent kinetic energy dissipation rate from low and moderate resolution velocity fluctuation time series"

_Atmospheric Measurement Techniques, 2016_

## Referee Comment (RC1) · Anonymous Referee #1 · 3 Apr 2017

**Manuscript ID** AMT-2016-401

**Title:** Novel approaches to estimating turbulent kinetic energy dissipation rate from low and moderate resolution velocity fluctuation time series

**Authors:** Marta Waclawczyk, Yong-Feng Ma, Jacek M. Kopeć, and Szymon P. Malinowski

**General comments:**

The authors propose and evaluate two techniques to retrieve estimates of the turbulent kinetic energy dissipation rate $\epsilon$ on the basis of the frequency of zero-crossings of time series of turbulent velocity fluctuations measured with sensors that do not have a frequency response that is necessary to resolve the smallest turbulent scales, which are on the order of the Kolmogorov length, $\eta$.

Section 2 ("State of the art") is a collection of equations that relate various turbulence characteristics with each other. These equations have been taken from a large number of different sources, and it is not clear what the underlying physical assumptions are and to what extent they are consistent across the various sources. For example, it is not explained whether the one-dimensional spectra $E_{11}(k_1)$ and $E_{22}(k_1)$ and the frequency spectrum $S(f)$ are meant to be one-sided spectra (such that the spectrum integrates to the variance when integrated from 0 to $\infty$) or two-sided spectra (such that the spectrum integrates to the variance when integrated from 0 to $\infty$). Moreover, it is not mentioned which of the relationships follow from Kolmogorov's theory of fully developed, locally homogeneous and isotropic turbulence Kolmogorov (1941a,b) and which are valid for any statistically homogeneous vector fields, regardless of whether or not they are isotropic (Monin and Yaglom, 1975, pp. 16-22).

Additionally, I find it worrisome that the authors sometimes confuse $k_1$ and $k$ and that some of their equations contain transcendental functions with dimensional arguments; see details in the specific comments below.

While the paper is in general well written, I find it difficult to follow the flow of the authors' reasoning in detail. I am not surprised that the zero-crossing methods can provide $\epsilon$ estimates with a quality comparable to the $\epsilon$ estimates obtained with traditional spectral retrieval methods. The relative advantages and disadvantages, however, are less clear, and the authors do not discuss and explain them in sufficient depth from a physical point of view.

**Recommendation:**

The paper may be acceptable for publication after major revision.

**Specific comments:**

1. Page 2, line 23: "State of the art" — This section heading is unnecessarily vague and misleading. I would suggest to replace it with "Previous methods to retrieve the energy dissipation rate from measured velocity time series" or something similar.

2. P. 3, lines 5ff: "The energy-spectrum function in the whole wavenumber range can be approximated by the formula (Pope, 2000) . . ." — It is not correct that Eq. (2) is an acceptable approximation of the energy spectrum for the *whole* wavenumber range. In particular, Eq. (2) fails at wave numbers small compared to $1/L$, where the turbulence is usually anisotropic and is no longer universal.

3. P. 3, lines 12ff.: "Within the validity of the Taylor's hypothesis (1) can be converted to the frequency spectra, where $k = (2\pi f)/U$ and $U$ is the mean velocity of the aircraft." — This statement is erroneous or misleading in two respects. First, Taylor's frozen-turbulence hypothesis converts the frequency $f$ to the longitudinal wave number $k_1$, not to the magnitude $k = \sqrt{k_1^2 + k_2^2 + k_3^2}$ of the three-dimensional wave vector $\mathbf{k}$. Second, $U$ is not the "mean velocity of the aircraft" but the magnitude of the vector difference between the aircraft velocity and the wind velocity. This magnitude is sometimes referred to as the "true air speed".

4. P. 7, lines 12-14: The integrals in Eqs. 25 and 26 contain the term $e^{-k}$. The exponential function, however, is a transcendental function and its argument must be dimensionless, such as the argument $\beta k \eta$ in Eq. 24. Because $k$ is not dimensionless (its dimension is 1/Length), Eqs. 25 and 26 cannot be correct.

**Bibliography**

Kolmogorov, A. N., 1941a: Local structure of turbulence in an incompressible fluid at very high Reynolds numbers. *Dokl. Akad. Nauk SSSR*, **30**, 299–303.

Kolmogorov, A. N., 1941b: Logarithmically normal distribution of the size of particles under fragmentation. *Dokl. Akad. Nauk SSSR*, **31**, 99–101.

Monin, A. S. and A. M. Yaglom, 1975: *Statistical fluid mechanics — Volume 2*. The MIT Press, Cambridge, Massachusetts, 874 pp.

---

## Author Comment (AC1) · 6 Apr 2017

We thank the Referee for the comments and suggestions. We will improve the manuscript accordingly. Below, we reply and discuss the issues raised up by the Referee.

**1   Reply to general comments**

1. *Section 2 ("State of the art") is a collection of equations that relate various turbulence characteristics with each other. These equations have been taken from*

*a large number of different sources, and it is not clear what the underlying phys-
ical assumptions are and to what extent they are consistent across the various
sources. For example, it is not explained whether the one-dimensional spectra
$E_{11}(k)$ and $E_{22}(k_1)$ and the frequency spectrum $S(f)$ are meant to be one-sided
spectra or two-sided spectra.*

$E_{11}(k)$, $E_{22}(k_1)$ and $S(f)$ are the one-sided spectra which, after integration over
argument from $0$ to $\infty$ equal the variance of the signal. $E_{ij}(k_1)$ are defined as
twice the one-dimensional Fourier transform of
$R_{ij}(r_1\mathbf{e}_1) = \langle u_i(\mathbf{x} + r_1\mathbf{e}_1, t)u_j(\mathbf{x}, t)\rangle$ (Pope, 2000). Here, we assume that the flow
is homogeneous and statistically stationary and statistics do not depend on point
$\mathbf{x}$ or time. Equations [11] in the manuscript will be corrected to

$$R_{11}(r_1\mathbf{e}_1) = \frac{1}{2}\int_{-\infty}^{\infty} E_{11}(k_1)\mathrm{e}^{ik_1r_1}\mathrm{d}k_1 = \int_0^{\infty} E_{11}(k_1)\cos\left(k_1r_1\right)\mathrm{d}k_1, \quad (1)$$

$$R_{11}''(r_1\mathbf{e}_1) = -\int_0^{\infty} E_{11}(k_1)k_1^2\cos\left(k_1r_1\right)\mathrm{d}k_1. \quad (2)$$

We will amend Section 2, correct Eqs. [11] and improve the manuscript in order
to make the considerations more consistent. (In this reply we refer to equations
from the manuscript using square brackets)

2. *Moreover, it is not mentioned which of the relationships follow from Kolmogorov's
theory of fully developed, locally homogeneous and isotropic turbulence Kol-
mogorov (1941a,b) and which are valid for any statistically homogeneous vector
fields, regardless of whether or not they are isotropic (Monin and Yaglom, 1975,
pp. 16-22)*

Eqs. [11] in the manuscript are valid under the assumption of homogeneity alone,
however, for further relations (relationship between $\lambda_g$ and $\lambda_f$ and Eqs. [15] and
[16]) the assumption of local isotropy (Kolmogorov, 1941) is needed, to finally find
the value of the dissipation rate.

As the airborne measurements provide signals of velocity along the $1D$ aircraft flight path, the local isotropy assumption is needed to estimate the dissipation rate $\epsilon$ of a 3D turbulent field and the assumption of homogeneity alone (Monin and Yaglom, 1975) is not sufficient.

3. *Additionally, I find it worrisome that the authors sometimes confuse $k_1$ and $k$ and that some of their equations contain transcendental functions with dimensional arguments.*

   Integration in equations [25], [26] and [31] is in fact performed over non-dimensional variables. In the manuscript we denoted them by $k$ and $k_1$ which was confusing. In "Reply to specific comments" below we present the derivation in detail and denote the variables by $\xi$ and $\xi_1$.

4. *I find it difficult to follow the flow of the authors' reasoning in detail. I am not surprised that the zero-crossing methods can provide $\epsilon$ estimates with a quality comparable to the $\epsilon$ estimates obtained with traditional spectral retrieval methods. The relative advantages and disadvantages, however, are less clear, and the authors do not discuss and explain them in sufficient depth from a physical point of view.*

   In the manuscript we proposed two extensions of zero-crossing method to estimate TKE dissipation rate for low-pass filtered signals, in particular from airborne turbulence measurements with spatial resolution of meters or tens of meters along $1D$ aircraft tracks. The first of them, described in Sections 3.1 and 4.1 applies additional filtering of the signal and, similarly as the structure function or power spectra methods, is based on the inertial-range arguments.

   In spite of the same underlying physical arguments the structure function and power spectra methods are often used simultaneously, for better $\epsilon$ estimates (Chamecki and Dias, 2004). Here, the proposed method offers yet another option.

The possible advantages are:

- simplicity (e.g. it is not necessary to choose averaging windows),
- robustness to measurement errors in recorded amplitude of velocity fluctuations (see discussion in conclusion section).

In all the three, listed above, methods based on the inertial range arguments it is necessary to use a constant $C_1$ or $C_2$, which has to be determined by independent measurements. In order to avoid this limitation we propose the second extension of zero crossings, described in Sections 3.2 and 4.2. This extension assumes a model spectrum for the inertial and the dissipation range. We apply the particular, exponential model, see Pope (2000). The advantage of this method is that the inertial-range constant $C_1$ cancels in Eq. [25] and the resulting value of $\epsilon$ is unaffected by the uncertainty in $C_1$ estimates. Clearly, the approach is as good as the model itself. The only parameter present in the model equations [25] and [26], $\beta$, is fixed by theoretical constrains, as the dissipation spectrum $2\nu k^2 E(k)$ should integrate to $\epsilon$. The second approach is based on different physical arguments than the methods based on the inertial-range scaling only, it additionally makes use of the first similarity hypothesis of Kolmogorov (1941) and a model for the dissipation range spectrum. Still, it can be used for signals with spectral cut-offs, hence it offers an alternative to the spectral retrieval methods.

**2 Reply to specific comments**

1. We will change the title of the section, according to the Referee's suggestion.

2. *Eq. [2] fails at wave numbers small compared to* $1/L$*, where the turbulence is usually anisotropic and is no longer universal.*

[Figure]

In principle, we agree here with the Referee. However, one can assume that the lowest wavenumbers of the spectrum $E_{11}$ available from the measurements are within the validity of the local isotropy assumption and that the largest scales of the flow do not influence the value of dissipation rate.

3. We agree with the Referee, we will amend the manuscript, accordingly.

4. *The integrals in Eqs. [25] and [26] contain the term* $e^{-k}$, *the exponential function, however, is a transcendental function and its argument must be dimensionless, such as the argument* $\beta k \eta$ *in Eq. [24]. Because* $k$ *is not dimensionless (its dimension is 1/Length), Eqs. [25] and [26] cannot be correct.*

Integration in Eqs. [25] and [26] is performed over non-dimensional variables. Below we present the derivation in detail and change the notation to $\xi$ and $\xi_1$ instead of $k$ and $k_1$.

In order to derive Eqs. [25] and [26] we considered a relation [22] between $N_{cut}$ and $N_L$

$$u'^2 N_L^2 = u_{cut}'^2 N_{cut}^2 \left( 1 + \frac{\int_{k_{cut}}^{\infty} k_1^2 E_{11} \mathrm{d}k_1}{\int_0^{k_{cut}} k_1^2 E_{11} \mathrm{d}k_1} \right), \tag{3}$$

and in Eq. [23] assumed a certain form of the energy spectrum applicable in the inertial and the dissipation range

$$E(k) = C \epsilon^{2/3} k^{-5/3} e^{-\beta k \eta}, \tag{4}$$

the corresponding one-dimensional spectrum $E_{11}$, for the range of scales where the local isotropy assumption holds, was calculated using Eq. [2] form the manuscript

$$E_{11}(k_1) = C \epsilon^{2/3} \int_{k_1}^{\infty} k^{-8/3} e^{-\beta k \eta} \left( 1 - \frac{k_1^2}{k^2} \right) \mathrm{d}k. \tag{5}$$

[Figure]

With the following change of variables $\xi = \beta k \eta$ in this integral we obtain

$$E_{11}(k_1) = C\epsilon^{2/3}(\beta\eta)^{5/3} \int_{k_1\beta\eta}^{\infty} \xi^{-8/3}\mathrm{e}^{-\xi}\left(1 - \frac{(k_1\beta\eta)^2}{\xi^2}\right)\mathrm{d}\xi. \tag{6}$$

We next introduce (6) into (3) and once again change the variables to $\xi_1 = k_1\beta\eta$. We obtain

$$u'^2 N_L^2 = u_{cut}'^2 N_{cut}^2 \left[1 + \frac{\int_{k_{cut}\beta\eta}^{\infty} \xi_1^2 \int_{\xi_1}^{\infty} \xi^{-8/3}\mathrm{e}^{-\xi}\left(1 - \frac{\xi_1^2}{\xi^2}\right)\mathrm{d}\xi\mathrm{d}\xi_1}{\int_0^{k_{cut}\beta\eta} \xi_1^2 \int_{\xi_1}^{\infty} \xi^{-8/3}\mathrm{e}^{-\xi}\left(1 - \frac{\xi_1^2}{\xi^2}\right)\mathrm{d}\xi\mathrm{d}\xi_1}\right] = u_{cut}'^2 N_{cut}^2 \mathcal{C}_{\mathcal{F}}, \tag{7}$$

where the correcting factor $\mathcal{C}_{\mathcal{F}}$ equals

$$\mathcal{C}_{\mathcal{F}} = 1 + \frac{\int_{k_{cut}\beta\eta}^{\infty} \xi_1^2 \int_{\xi_1}^{\infty} \xi^{-8/3}\mathrm{e}^{-\xi}\left(1 - \frac{\xi_1^2}{\xi^2}\right)\mathrm{d}\xi\mathrm{d}\xi_1}{\int_0^{k_{cut}\beta\eta} \xi_1^2 \int_{\xi_1}^{\infty} \xi^{-8/3}\mathrm{e}^{-\xi}\left(1 - \frac{\xi_1^2}{\xi^2}\right)\mathrm{d}\xi\mathrm{d}\xi_1}. \tag{8}$$

The form of both equations is identical as [25] and [26], but the integration variables are denoted $\xi$ and $\xi_1$, instead of $k$ and $k_1$. We will amend the manuscript, accordingly. At the same time we note that the results of analysis performed in Sections 4 and 5 remain unchanged.

**References**

M. Chamecki and N. L. Dias, 2004, The local isotropy hypothesis and the turbulent kinetic energy dissipation rate in the atmospheric surface layer, Q. J. R. Meteorol. Soc., **130**, pp. 2733-2752

A. N. Kolmogorov, 1941, Local structure of turbulence in an incompressible fluid at very high Reynolds numbers, *Sov. Phys. Usp.*, **10**, 734-736, *reprinted from Dokl. Akad. Nauk SSSR*, **30**, 299-303.

A. S. Monin and A. M. Yaglom, 1975, Statistical fluid mechanics, Vol. 2, The MIT Press, Cambridge, Massachusetts

S. B. Pope, 2000, *Turbulent flows*, Cambridge.

---

## Referee Comment (RC2) · Anonymous Referee #2 · 9 May 2017

I was conflicted in how to approach my review: on one hand, there are no glaring technical problems in the authors approach, but on the other hand, they clearly do not make the case for any practical use of their method. Specifically, their approach is highly dependent on the form of the power spectrum in the dissipation range, and yet in their application there are no data points in that range (i.e., all the points are in the inertial subrange). So, why would one want to use their method - with no actual data in the dissipation range, and a potentially suspect model in that range - over a more-standard method based on a tried-and-true inertial subrange model - and where most of their data lie. They do not perform a sensitivity study on the choice of dissipation range model. They use a specific exponential model from Pope (2000), but if they

had read the discussion in that reference, they would have noted that Pope does not consider that model to be accurate. And as the authors point out, the dissipation range spectrum has a significant effect on the number of zero crossings. Furthermore, they do not address practical issues inherent in digital signal processing: spectral bias due to finite temporal windows, aliasing due to temporal sampling, as well as sensor bias and noise. It seems that these artifacts might be have a significant impact on a zero-crossing method. For example, it is not hard to see how sensor bias and noise, could significantly impact zero crossings, especially for low SNR data.

So, they need to address the question of why one would want to use their method over more standard approaches (unless of course, one had data with significant content in the dissipation range), and how their method is susceptible/tolerant to signal processing artifacts. I feel strongly that they need to perform a simulation analysis to answer these questions in a statistical sense (see for example, Frehlich, et al. JAM 2001); real data is not acceptable, except for case studies. As the paper stands, I would require significant revisions that address these issues, before accepting for publication.
* * *

---

## Author Comment (AC2) · 6 Jun 2017

We would like to thank the Referee for the comments and suggestions. Below, we reply and discuss the issues raised up by the Referee and present the planned amendments of the manuscript.

1. *They do not perform a sensitivity study on the choice of dissipation range model. They use a specific exponential model from Pope (2000), but if they had read the discussion in that reference, they would have noted that Pope does not consider that model to be accurate. And as the authors point out, the dissipation range*

[Figure]

**Fig. 1.** Functions $E_{11}(k_1)$ and $k_1^2 E_{11}(k_1)$ calculated for the measured signal (black line), exponential $f_\eta$ (dot-dashed blue line), Pope spectrum (dashed magenta line).

*spectrum has a significant effect on the number of zero crossings.*

In the cited reference [Pope, 2000] three different forms for the function $f_\eta$ were considered, the exponential, the Pao and an improved form, which will be further referred to as the "Pope spectrum", see Eqs. (6.248), (6.249), (6.254) therein. All the three forms of the dissipative spectra integrate to $\epsilon$ i.e. they satisfy the requirement $\epsilon = 2\nu \int k^2 E(k)k$. According to the analysis of experimental data, the Pope spectrum provides the best fit in the dissipative range [Pope 2000].

In the revised manuscript we will compare results with both the exponential and the Pope forms of function $f_\eta$. We will show that the obtained $\epsilon$ estimates are very close to each other. To explain this we first note that in the proposed model (Eq. [22] in the manuscript) only the integral of the dissipative spectrum $k_1^2 E_{11}(k_1)$ is present. The spectral cut-off of the data considered in our work ($5Hz$) is in the inertial range, where $k_1^2 E_{11}(k_1)$ with both forms of $f_\eta$ functions are almost indistinguishable, see the attached figures (dashed magenta lines are for the Pope spectrum, dot-dashed blue lines for exponential $f_\eta$). At the same time integrals of the remaining (recovered) parts of $k_1^2 E_{11}(k_1)$ are almost equal (as both dissipative spectra $2\nu k^2 E(k)$ integrate to $\epsilon$). As a result, for the given spectral cut-off the $\epsilon$ estimates are almost the same, independently of the form of the $f_\eta$ function. This might change for larger cut-off frequencies. We expect that in case the cut-off frequency is placed in a region influenced by the form of $f_\eta$ function, the Pope spectrum will provide better estimates of the TKE dissipation rate. We will include the new results and the above discussion in the revised manuscript.

2. *Furthermore, they do not address practical issues inherent in digital signal processing: spectral bias due to finite temporal windows, aliasing due to temporal*

*sampling, as well as sensor bias and noise. It seems that these artefacts might have a significant impact on a zero- crossing method. For example, it is not hard to see how sensor bias and noise, could significantly impact zero crossings, especially for low SNR data.*

As suggested by the Referee we performed simulation analysis, [Frehlich et al. (2001), Sharman et al. (2014)], in order to address the issues of the influence of finite temporal windows and aliasing on $\epsilon$ estimates. In the revised manuscript we will present and discuss the obtained results. As far as the sensor bias is concerned, in fact both the variance of the noise as well as variance of its derivative influence the measured number of crossings. This issue was studied in detail by Sreenivasan et al. (1983), hence, we did not discussed it in the manuscript. Moreover, Poggi & Katul (2010) suggested to use the threshold- instead of the zero-crossings in case of low SNR signals. In our application the noise influences largely the higher frequencies (above $5Hz$) which are removed by the low-pass filter used in the proposed number of crossings method. Moreover, use of the threshold- instead of zero-crossings did not lead to any systematic change of our estimates. In the revised manuscript we will include a discussion concerning the sensor bias, referring to the two mentioned papers.

3. *So, they need to address the question of why one would want to use their method over more standard approaches (unless of course, one had data with significant content in the dissipation range), and how their method is susceptible/tolerant to signal processing artifacts. I feel strongly that they need to perform a simulation analysis to answer these questions in a statistical sense (see for example, Frehlich, et al. JAM 2001).*

Based on the results of the performed simulation analysis we will argue that the number of crossing method has certain advantages over standard methods. We created sets of artificial velocity signals with a prescribed form of the energy spectrum [Frehlich et al. (2001), Sharman et al. (2014)]. At least for these artificial

velocity signals, the obtained $\epsilon$ values were less sensitive to the aliasing error than the estimates from the power spectral method. Moreover, the bias due to the finite temporal windows was smaller for the number of crossing method, however, on the cost of larger uncertainty (larger standard deviations) of the estimated dissipation rate values.

These differences in errors of the number of crossing and the power spectral method can make the former an additional tool to improve $\epsilon$ estimates from the atmospheric measurements.

Moreover, we argue that the number of crossings method applied to the fully-resolved signals has become a fairly standard tool for $\epsilon$ estimates, used also in the atmospheric measurements, see e.g. Poggi & Katul (2010). Therein, the discussed advantages of the method are that no gradient measurements are required (to estimate the Taylor microscale $\lambda$), no assumptions about scaling laws in structure functions (and power spectra) are needed and no simplifications in the TKE budget are adopted (for which $\epsilon$ is computed as residual). The method proposed in the current manuscript, in particular, the second approach based on the recovered part of the spectrum, generalises number of crossing method and makes it applicable also for signals with spectral cut-off. Off course, an additional cost is that certain form of the energy spectrum must be assumed. The method can be interesting in particular for data with cut-offs reaching the dissipation range, but still with part of this range missing (or contaminated with noise). In such case, using only the inertial-range estimates may lead to a significant loss of information, as the data from the dissipation range are not taken into account.

**References**

Frehlich, R., Cornman, L., and Sharman R.: Simulation of Three-Dimensional Turbulent Velocity Fields, Journal of Applied Meteorology, 40, 246-258, 2001.
Poggi, D. and Katul, G. G.: Evaluation of the Turbulent Kinetic Energy Dissipation Rate Inside Canopies by Zero- and Level-Crossing Density Methods, Boundary-Layer Meteorol., 136, 219–233, 2010

Pope, S. B. , *Turbulent flows*, Cambridge, 2000.

Sharman, R. D., Cornman, L. B., Meymaris, G., Pearson, J. P., and Farrar, T.: Description and Derived Climatologies of Automated In Situ Eddy-Dissipation-Rate Reports of Atmospheric Turbulence, Journal of Applied Meteorology and Climatology, 53, 1416-1432.

Sreenivasan, K., Prabhu, A. and Narasimha, R.: Zero-crossings in turbulent signals, Journal of Fluid Mechanics, 137, 251–272, 1983.

─────────────────────────────

[Figure]

[Figure]

**Fig. 2.**

**Fig. 3.**

[Figure]

---

## Author Response (AR1)

We thank the Referees for the comments and suggestions. We have improved the manuscript accordingly. Below, we reply and discuss the issues raised up by the Referee.

**I. REPLY TO COMMENTS OF REFEREE 1**

1. *Section 2 ("State of the art") is a collection of equations that relate various turbulence characteristics with each other. These equations have been taken from a large number of different sources, and it is not clear what the underlying physical assumptions are and to what extent they are consistent across the various sources. For example, it is not explained whether the one-dimensional spectra $E_{11}(k)$ and $E_{22}(k_1)$ and the frequency spectrum $S(f)$ are meant to be one-sided spectra or two-sided spectra.*

   $E_{11}(k)$, $E_{22}(k_1)$ and $S(f)$ are the one-sided spectra which, after integration over argument from 0 to $\infty$ equal the variance of the signal. $E_{ij}(k_1)$ are defined as twice the one-dimensional Fourier transform of $R_{ij}(r_1\mathbf{e}_1) = \langle u_i(\mathbf{x},t)u_j(\mathbf{x}+r_1\mathbf{e}_1,t)\rangle$ (Pope, 2000). We assume that the flow is statistically stationary and statistics do not depend on time. We ammended Section 2, Eqs. [11] in the manuscript were be corrected to

$$R_{11}(r_1\mathbf{e}_1) = \int_0^\infty E_{11}(k_1)\cos(k_1 r_1)\mathrm{d}k_1, \tag{1}$$

$$R_{11}''(r_1\mathbf{e}_1) = -\int_0^\infty E_{11}(k_1)k_1^2\cos(k_1 r_1)\mathrm{d}k_1. \tag{2}$$

2. *Moreover, it is not mentioned which of the relationships follow from Kolmogorov's theory of fully developed, locally homogeneous and isotropic turbulence Kolmogorov (1941a,b) and which are valid for any statistically homogeneous vector fields, regardless of whether or not they are isotropic (Monin and Yaglom, 1975, pp. 16-22)*

   Eqs. [11-14] in the manuscript are valid under the assumption of homogeneity alone, however, for further relations (relationship between $\lambda_g$ and $\lambda_f$ and Eqs. [15] and [16]) the assumption of local isotropy (Kolmogorov, 1941) is needed, to finally find the value of the dissipation rate.

   As the airborne measurements provide signals of velocity along the $1D$ aircraft flight path, the local isotropy assumption is needed to estimate the dissipation rate $\epsilon$ of a 3D turbulent field and the assumption of homogeneity alone (Monin and Yaglom, 1975) is not sufficient.

We added this discussion to Section 2, page 5, lines 8-11.

3. *Additionally, I find it worrisome that the authors sometimes confuse $k_1$ and $k$ and that some of their equations contain transcendental functions with dimensional arguments...*

Integration in equations [25], [26] and [31] in the old manuscript was performed over non-dimensional variables $k\beta\eta$ and $k_1\beta\eta$. In the manuscript we denoted them by $k$ and $k_1$ which was confusing. In the ammended version we denoted the non-dimensional variables by $\xi$ and $\xi_1$, see Eqs. [27] and [38] in the new manuscript and added a sentence with details of derivation of Eq. [27].

4. *I find it difficult to follow the flow of the authors' reasoning in detail. I am not surprised that the zero-crossing methods can provide $\epsilon$ estimates with a quality comparable to the $\epsilon$ estimates obtained with traditional spectral retrieval methods. The relative advantages and disadvantages, however, are less clear, and the authors do not discuss and explain them in sufficient depth from a physical point of view.*

In the manuscript we proposed two extensions of zero-crossing method to estimate TKE dissipation rate for low-pass filtered signals, in particular from airborne turbulence measurements with spatial resolution of meters or tens of meters along $1D$ aircraft tracks. The first of them, described in Sections 3.1 and 4.1 applies additional filtering of the signal and, similarly as the structure function or power spectra methods, is based on the inertial-range arguments.

In spite of the same underlying physical arguments the structure function and power spectra methods are often used simultaneously, for better $\epsilon$ estimates (Chamecki and Dias, 2004). Here, the proposed method offers yet another option. Moreover, the second proposed method assumes a model spectrum for the inertial and the dissipation range, hence it is based on different physical arguments than the methods based on the inertial-range scaling only, it additionally makes use of the first similarity hypothesis of Kolmogorov (1941) and a model for the dissipation range spectrum. Still, it can be used for signals with spectral cut-offs, hence it offers an alternative to the spectral retrieval methods.

The possible advantages of the newly proposed approaches (apart from the simplicity

of the number of crossings detection) follow from the simulation analysis performed according to the suggestion of Referee 2. Results are presented in Section 4.2, it seems that at least for the generated synthetic turbulent signals, the number of crossings method is less sensitive to the bias error and an error due to aliasing than the spectral retrieval method.

Advantages and disadvantages of the new approaches are now discussed in more detail in Conclusions.

5. *Page 2, line 23: State of the art  This section heading is unnecessarily vague and mislead- ing. I would suggest to replace it with Previous methods to retrieve the energy dissipation rate from measured velocity time series or something similar*

We changed the title of the section, according to the Referee's suggestion.

6. *Eq. [2] fails at wave numbers small compared to $1/L$, where the turbulence is usually anisotropic and is no longer universal.*

In principle, we agree here with the Referee. However, one can assume that the lowest wavenumbers of the spectrum $E_{11}$ available from the measurements are within the validity of the local isotropy assumption and that the largest scales of the flow do not influence the value of dissipation rate. In the revised manuscript on page 3, text before Eq. (3) now reads: "Within the validity of the local isotropy assumption of Kolmogorov (1941), the energy-spectrum function can be approximated by the formula (Pope, 2000):"

7. *P. 3, lines 12ff.: Within the validity of the Taylors hypothesis (1) can be converted to the frequency spectra, where $k = (2f)/U$ and $U$ is the mean velocity of the aircraft. This statement is erroneous or misleading in two respects. First, Taylors frozen-turbulence hypothesis converts the frequency $f$ to the longitudinal wave number $k_1$, not to the magnitude $k = |\mathbf{k}|$ of the three-dimensional wave vector $\mathbf{k}$. Second, $U$ is not the mean velocity of the aircraft but the magnitude of the vector difference between the aircraft velocity and the wind velocity. This magnitude is sometimes referred to as the true air speed.*

We agree with the Referee, we amended the manuscript, accordingly (see page 3 line 12).

8. *The integrals in Eqs. 25 and 26 contain the term* $e^{-k}$*, the exponential function, how-ever, is a transcendental function and its argument must be dimensionless, such as the argument* $\beta k\eta$ *in Eq. [24]. Because* $k$ *is not dimensionless (its dimension is 1/Length), Eqs. [25] and [26] cannot be correct.*

Integration in Eqs. [25] and [26] in the old manuscript was performed over non-dimensional variables. Below we present the derivation in detail and change the notation to $\xi$ and $\xi_1$ instead of $k$ and $k_1$.

In order to derive Eqs. [25] and [26] we considered a relation between $N_{cut}$ and $N_L$

$$u'^2 N_L^2 = u'^2_{cut} N^2_{cut} \left( 1 + \frac{\int_{k_{cut}}^{\infty} k_1^2 E_{11} dk_1}{\int_0^{k_{cut}} k_1^2 E_{11} dk_1} \right), \tag{3}$$

and assumed a certain form of the energy spectrum applicable in the inertial and the dissipation range

$$E(k) = C\epsilon^{2/3} k^{-5/3} e^{-\beta k\eta}, \tag{4}$$

the corresponding one-dimensional spectrum $E_{11}$, for the range of scales where the local isotropy assumption holds, was calculated using Eq. [2] form the manuscript

$$E_{11}(k_1) = C\epsilon^{2/3} \int_{k_1}^{\infty} k^{-8/3} e^{-\beta k\eta} \left( 1 - \frac{k_1^2}{k^2} \right) dk. \tag{5}$$

With the following change of variables $\xi = \beta k\eta$ in this integral we obtain

$$E_{11}(k_1) = C\epsilon^{2/3} (\beta\eta)^{5/3} \int_{k_1\beta\eta}^{\infty} \xi^{-8/3} e^{-\xi} \left( 1 - \frac{(k_1\beta\eta)^2}{\xi^2} \right) d\xi. \tag{6}$$

We next introduce (6) into (3) and once again change the variables to $\xi_1 = k_1\beta\eta$. We obtain

$$u'^2 N_L^2 = u'^2_{cut} N^2_{cut} \left[ 1 + \frac{\int_{k_{cut}\beta\eta}^{\infty} \xi_1^2 \int_{\xi_1}^{\infty} \xi^{-8/3} e^{-\xi} \left( 1 - \frac{\xi_1^2}{\xi^2} \right) d\xi d\xi_1}{\int_0^{k_{cut}\beta\eta} \xi_1^2 \int_{\xi_1}^{\infty} \xi^{-8/3} e^{-\xi} \left( 1 - \frac{\xi_1^2}{\xi^2} \right) d\xi d\xi_1} \right] = u'^2_{cut} N^2_{cut} \mathcal{C}_{\mathcal{F}}, \tag{7}$$

where the correcting factor $\mathcal{C}_{\mathcal{F}}$ equals

$$\mathcal{C}_{\mathcal{F}} = 1 + \frac{\int_{k_{cut}\beta\eta}^{\infty} \xi_1^2 \int_{\xi_1}^{\infty} \xi^{-8/3} e^{-\xi} \left( 1 - \frac{\xi_1^2}{\xi^2} \right) d\xi d\xi_1}{\int_0^{k_{cut}\beta\eta} \xi_1^2 \int_{\xi_1}^{\infty} \xi^{-8/3} e^{-\xi} \left( 1 - \frac{\xi_1^2}{\xi^2} \right) d\xi d\xi_1}. \tag{8}$$

The form of both equations is identical as [25] and [26], but the integration variables are denoted $\xi$ and $\xi_1$, instead of $k$ and $k_1$. We amended the manuscript, accordingly. At the same time we note that the results of analysis performed in Sections 4 and 5 remained unchanged.

**II. REPLY TO COMMENTS OF REFEREE 2**

1. *They do not perform a sensitivity study on the choice of dissipation range model. They use a specific exponential model from Pope (2000), but if they had read the discussion in that reference, they would have noted that Pope does not consider that model to be accurate. And as the authors point out, the dissipation range spectrum has a significant effect on the number of zero crossings.*

   In the cited reference [Pope, 2000] three different forms for the function $f_\eta$ were considered, the exponential, the Pao and an improved form, which will be further referred to as the "Pope spectrum", see Eqs. (6.248), (6.249), (6.254) therein. All the three forms of the dissipative spectra integrate to $\epsilon$ i.e. they satisfy the requirement $\epsilon = 2\nu \int k^2 E(k)\mathrm{d}k$. According to the analysis of experimental data, the Pope spectrum provides the best fit in the dissipative range [Pope 2000].

   In the revised manuscript, in Section 4.3 we compared results with both the exponential and the Pope forms of function $f_\eta$. We show that the obtained $\epsilon$ estimates are very close to each other. To explain this we first note that in the proposed model (Eq. [22] in the manuscript) only the integral of the dissipative spectrum $k_1^2 E_{11}(k_1)$ is present. The spectral cut-off of the data considered in our work $(5Hz)$ is in the inertial range, where $k_1^2 E_{11}(k_1)$ with both forms of $f_\eta$ functions are almost indistinguishable, see Fig. 1. At the same time integrals of the remaining (recovered) parts of $k_1^2 E_{11}(k_1)$ are almost equal (as both dissipative spectra $2\nu k^2 E(k)$ integrate to $\epsilon$). As a result, for the given spectral cut-off the $\epsilon$ estimates are almost the same, independently of the form of the $f_\eta$ function. This might change for larger cut-off frequencies. We expect that in case the cut-off frequency is placed in a region influenced by the form of $f_\eta$ function, the Pope spectrum will provide better estimates of the TKE dissipation rate. We included the new results and the above discussion in the revised manuscript in Section 4.3.

2. *Furthermore, they do not address practical issues inherent in digital signal processing: spectral bias due to finite temporal windows, aliasing due to temporal sampling, as well as sensor bias and noise. It seems that these artefacts might have a significant impact on a zero- crossing method. For example, it is not hard to see how sensor bias and*

[Figure]

[Figure]

FIG. 1. Functions $E_{11}(k_1)$ and $k_1^2 E_{11}(k_1)$ calculated for the measured signal (black line), exponential spectrum (dot-dashed blue line), Pope spectrum (dashed magenta line).

*noise, could significantly impact zero crossings, especially for low SNR data.*

As suggested by the Referee we performed simulation analysis, [Frehlich et al. (2001), Sharman et al. (2014)], in order to address the issues of the influence of finite temporal windows and aliasing on $\epsilon$ estimates. In the revised manuscript, in Section 4.2, we present and discuss the obtained results. As far as the sensor bias is concerned, in fact both the variance of the noise as well as variance of its derivative influence the measured number of crossings. This issue was studied in detail by Sreenivasan et al. (1983), hence, we did not discussed it in the manuscript. Moreover, Poggi & Katul (2010) suggested to use the threshold- instead of the zero-crossings in case of low SNR signals. In our application the noise influences largely the higher frequencies (above $5Hz$) which are removed by the low-pass filter used in the proposed number of crossings method. Moreover, use of the threshold- instead of zero-crossings did not lead to any systematic change of our estimates. In the revised manuscript we included a discussion concerning the sensor bias, referring to the two mentioned papers, in Section 4.2.

3. *So, they need to address the question of why one would want to use their method over more standard approaches (unless of course, one had data with significant content in the dissipation range), and how their method is susceptible/tolerant to signal processing*

*artifacts. I feel strongly that they need to perform a simulation analysis to answer these questions in a statistical sense (see for example, Frehlich, et al. JAM 2001).*

Based on the results of the performed simulation analysis we argue that the number of crossing method has certain advantages over standard methods. We created sets of artificial velocity signals with a prescribed form of the energy spectrum [Frehlich et al. (2001), Sharman et al. (2014)]. Results are presented in Section 4.2. At least for these artificial velocity signals, the obtained $\epsilon$ values were less sensitive to the aliasing error than the estimates from the power spectral method. Moreover, the bias due to the finite temporal windows was smaller for the number of crossing method, however, on the cost of larger uncertainty (larger standard deviations) of the estimated dissipation rate values.

These differences in errors of the number of crossing and the power spectral method can make the former an additional tool to improve $\epsilon$ estimates from the atmospheric measurements.

Moreover, we argue that the number of crossings method applied to the fully-resolved signals has become a fairly standard tool for $\epsilon$ estimates, used also in the atmospheric measurements, see e.g. Poggi & Katul (2010). Therein, the discussed advantages of the method are that no gradient measurements are required (to estimate the Taylor microscale $\lambda$), no assumptions about scaling laws in structure functions (and power spectra) are needed and no simplifications in the TKE budget are adopted (for which $\epsilon$ is computed as residual). The method proposed in the current manuscript, in particular, the second approach based on the recovered part of the spectrum, generalises number of crossing method and makes it applicable also for signals with spectral cut-off. Off course, an additional cost is that certain form of the energy spectrum must be assumed. The method can be interesting in particular for data with cut-offs reaching the dissipation range, but still with part of this range missing (or contaminated with noise). In such case, using only the inertial-range estimates may lead to a significant loss of information, as the data from the dissipation range are not taken into account.

In the revised manuscript we extended the discussion about advantages and disadvantages of the new methods in Conclusions.

**III.  LIST OF CHANGES**

1. We modified Section 2, according to the suggestions of Referee 1, we described the applied assumptions, corrected Eqs. [11] and in Eq. [4] we use constant $C_1$ instead of $\alpha$. We corrected the text on page 3, line 13 to $k_1 = (2\pi f)/U$ and defined $U$ according to the suggestion of Referee 1.

2. In Eq. [9] we use $k_1$ instead of $k$

3. We corrected the definition of $R_{11}$ on page 4, line 23. We now write "We will now introduce the two-point correlation of velocity $R_{ij}(r_1\mathbf{e}_1) = \langle u_i(\mathbf{x},t)u_j(\mathbf{x}+r_1\mathbf{e}_1,t)\rangle$ and assume that the flow is statistically stationary and statistics do not depend on time."

4. According to suggestions of Referee 2 in Section 3.2 we discuss both the simple exponential and the Pope spectrum in Eq. (23), (24) in the following formulas instead of $e^{\beta k \eta}$ we write $f_\eta(\beta k \eta)$

5. We added Section 4.2 "Simulation analysis and error estimates" according to suggestions of Referee 2.

6. In Section 4.3 we added result of $\epsilon$ estimates with the use of the Pope spectrum and a discussion of results.

7. In conclusions we discuss advanteges and disadvantages of new proposals in more detail, as suggested by both Referees.

8. We replaced Fig. 13 with results obtained from more precise calculations of integrals on non-uniform grid.
* * *

[revised manuscript text omitted]

---

## Referee Report (RR1)

[referee-annotated manuscript omitted]

---

## Author Response (AR2)

Reply to Referee:

The authors would like to thank the Referee for the comments and suggestions. Below we present our detailed reply and discussion.

General comments:

1. "This issue is most likely the reason why the "psd" results are biased. The correct formulas to use are in two of your references (Frehlich et al. and Sharman et al.) including a maximum likelihood (ML) EDR estimation method from the latter reference."

"T*he glaring issue is that when you calculate the energy dissipation rate (EDR) from the simulated data, you are not using the correct model spectrum. When sampling and using a finite window's worth of data, the average power spectrum will be the expected periodogram, not the theoretical spectrum (in this case the "-5/3rds" one). This is most likely the reason your "psd" results are biased. Therefore, it is anticipated correction will show that your method - in its current form - is inferior to existing method*"

In the revised version that we sent on the 6$^{th}$ of June we addressed the previous comment of the Referee (sent on the 9$^{th}$ May):  "Furthermore, they do not address practical issues inherent in digital signal processing: spectral bias due
to finite temporal windows, aliasing due to temporal sampling, as well as sensor bias and noise."
Hence, we aimed to present the sensitivity of the results on the different types of error, without using any corrections. In the suggested reference, instead of the theoretical von K\'arm\'an model, "the periodogram of the computed windowed von K\'arm\'an autocorrelation function" is ued. This function already accounts for the bias errors, hence, one cannot expect the "uncorrected" number of crossing to perform better than the bias-corrected spectral methods.  Moreover, as written by Sharman et al.  "In the case of commercial aircraft the details of the filtering are often not known, and the empirical parameter $\gamma$ in Eq. (16) is used to account for these effects." Hence, using the peridogram for a {\it  a priori} known filter does not provide a universal solution to the problem.

We argue, analogous bias corrections as presented in  Sharman et al. could be proposed for the number of crossing methods. However, the aim of our manuscript was to introduce the number of crossing aproaches for signals with spectral cut-offs. We also addressed, as the Reviewer requested in the first review, different types of errors which may influence the results. We agree that the issue of bias-correction can be important, however,  it is beyond the scope of the present paper.

In the second revision we addressed the method presented in Sharman et al. and clearly pointed out that it accounts for the effects of the filtering window.

2. "*The correction will be very important in that the results of your simulations now show bias in the standard psd approach, which will expectedly be corrected. It is noted that this standard approach has already much less scatter than the new suggested method. (See for example, Fig 8.)*"

In fact, as seen in Fig. 8 of the manuscript, the method based on the number of crossings has a larger scatter than the psd method, at least for the chosen range of filter cut-offs. We

investigated this problem in order to address the Referee objections. It follows from our study that the scatter in $\epsilon_{NCF}$ depends on the value of filter cut-offs In the fitting range. In the revised version we compare results for short signals (from $2^8$ points) and short fitting range $f=[16\ 18]$ using the number of crossings and power spectrum methods. We found the standard deviation of $\epsilon_{NCF}$ comparable with the psd method.

The detected number of crossings is larger for higher cut-offs. Hence, especially for the case of short signals, the statistics are reproduced better if larger $f_{cut}$'s are considered.

3. "*Therefore, it is anticipated correction will show that your method - in its current form - is inferior to existing methods.*"

We find such statement unfair. The EDR retrieval methods based on the power spectral density were investigated in numerous works, including the suggested reference of Sharman et al. We proposed, for the first time, alternative approaches for signals with moderate resolutions, based on the number of crossings and it was the main subject of the manuscript. Moreover, we showed that the new method responds differently than psd method to errors introduced by the filter, which in principle is an advantage – two methods used in parallel give better understanding of possible imperfections of EDR retrievals. In the revised version we also address the issue of a larger scatter, which, for short signals and short fitting ranges depends on the $f_{cut}$ values in the fitting range. The number of crossings statistics are calculated with a higher accuracy for higher $f_{cut}$ values.

In our opinion such results can increase the number of possible future investigations and better retrievals of EDR.

Detailed changes:

page 1, line 23: We wrote "frequencies" as we referred to the measured time-series. In the new version we write:

"Using the Taylor's hypothesis, the measured time series can be converted into a spatial signal and the sampling frequency will correspond to scales which are typically $2-3$ orders of magnitude larger than the Kolmogorov scales."

page 3, line 1: we wrote that $k_1$ is measured in $rad/m$.
line 30: instead of "stationary signal" we write "homogeneous velocity signal, converted to time series $u(t)$ with the use of Taylor's hypothesis."

page 4, line 3: In Section 2 we addressed previous EDR retrieval methods. We only referred to proposal of Fairall et al., 1980; where filter effects were not accounted for. We wrote "Assuming that the filter is perfect, i.e. it is a rectangle in the frequency space, after the filtering..." As for the method proposed in the manuscript, we discuss the issue of frequency response characteristics in Section 4.1.

page 5, line 12: We defined $k_c$ k c as the characteristic wavenumber along the longitudinal direction

page 10, line 5-10: We address the Referee's objection about the Gaussianity of the signal. In fact, according to the work of Rice (1945) the Gaussianity of the signal itself is a necessary, but not a sufficient condition. However, as follows from the study of

Sreenivasan et al. (1983), the Rice formula was satisfied with a good accuracy even for strongly non-Gaussian pdf's of a signal and its derivative.

Page 10, line 12: We address the Referee comment "but how does it potentially effect your results? Your cutoff frequencies go above 10 Hz, so how can the results for those cases be acceptable?"

For the signals from POST we use cut-off frequencies up to $5$ Hz. We wrote:
"However, as the highest cut-off frequencies used in the present study are 5 Hz, it should not affect our results."

Page 11, lines 10-15: We reformulated the beginning of chapter 4.2, we write that the signal-to-noise ratio becomes significant at higher frequencies.

We reformulated Section 4.2.

Page 20, line 3 and 7. As we changed the method of calculating integrals in Eq. (35), using non-uniform grids, results for $\epsilon_{NCR}$ changed (were improved).

We reformulated conclusions. We note that when writing about advantages of the original number of crossing method we referred to the work of Poggi and Katul (2010) where this method was used for EDR estimation inside canopies.

---

## Author Response (AR3)

We would like to thank the Editor for his remarks and comments. We ammended the manuscript accordingly.

1. We extended the abstract adding the following sentences about results:

[revised manuscript text omitted]